# Bayesian interval estimations for the mean of delta-three parameter lognormal distribution with application to heavy rainfall data

**Patcharee Maneerat[1], Pisit Nakjai[2], Sa-Aat Niwitpong[3]***

**1** Department of Applied Mathematics, Uttaradit Rajabhat University, Uttaradit, Thailand, **2** Department of Computer Sciences, Uttaradit Rajabhat University, Uttaradit, Thailand, **3** Department of Applied Statistics, King Mongkut's University of Technology North Bangkok, Bangkok, Thailand

* sa-aat.n@sci.kmutnb.ac.th

**Data Availability Statement:** All relevant data are within the manuscript and its Supporting information files.

## Abstract

Flash flooding is caused by heavy rainfall that frequently occurs during a tropical storm, and the Thai population has been subjected to this problem for a long time. The key to solving this problem by planning and taking action to protect the population and infrastructure is the motivation behind this study. The average weekly rainfall in northern Thailand during Tropical Storm Wipha are approximated using interval estimations for the mean of a delta-three parameter lognormal distribution. Our proposed methods are Bayesian confidence intervals-based noninformative (NI) priors (equal-tailed and highest posterior density (HPD) intervals based on NI1 and NI2 priors). Our numerical evaluation shows that the HPD-NI1 prior was closer to the nominal confidence level and possessed the narrowest expected length when the variance was small-to-medium for a large threshold. The efficacy of the methods was illustrated by applying them to weekly natural rainfall data in northern Thailand to examine their abilities to indicate flooding occurrence.

## Introduction

Human beings and all living things need water to survive, so life would not exist without it. The amount of water that is usually available depends on the amounts of rain and snow that fall. Unfortunately, some areas barely see rain while others get more than their fair share. These situations can cause natural disasters such as floods and droughts, which are dramatic changes that occur when most people least expect them. In Thailand, long periods of rain caused by tropical storms have triggered significant flooding. In July-August 2019, Tropical Storm Wipha crossed the North Vietnam coast and headed westward toward upper Thailand (mainly the northern and northeastern regions). Heavy rain passing through northern Thailand caused flash flooding in Phayao province and also triggered landslides in Nan province in northern Thailand [1]. As a consequence, these extreme events caused loss of life and significantly damaged assets and transport infrastructure in these at-risk areas. One of the most important factors in solving this problem is how to use the historical rainfall data to plan and prevent more flooding in the future by taking direct action accordingly. These reasons have

**Funding:** This research was funded by King Mongkut's University of Technology North Bangkok, Contract no. KMUTBNB-65-KNOW-09. Dr. Patcharee Maneerat and Dr. Pisit Nakjai were appreciated funding by Thailand Science Research and Innovation (TSRI) and Uttaradit Rajabhat University. The funders had no role in study design, data collection and analysis, decision to publish, or preparation of the manuscript.

**Competing interests:** The authors have declared that no competing interests exist.

led to our interest in the estimation of the mean rainfall amount using the observed data from extreme rainfall events. Importantly, the weekly natural rainfall amounts in the week 29 July to 4 August 2019 follow the assumptions of a delta-three parameter lognormal distribution.

A three parameter lognormal distribution is considered to be suitable for the observed data that are highly skewed and cannot be modeled using a two-parameter lognormal distribution [2]. A delta-three parameter lognormal distribution is a combination of zero and non-zero values following the three-parameter lognormal distribution first introduced by Aitchison and Brown [3]. The three-parameter lognormal distribution has threshold parameter $a$, an unknown parameter that makes it differ from a two-parameter lognormal distribution in that $a > 0$. Thus, a two-parameter lognormal distribution is a special case of a three-parameter lognormal distribution when $a = 0$. In application involving real-world data, the three-parameter lognormal distribution has been used in Hydrology [4–6], rainfall network [7, 8], and flood frequency analysis [9, 10].

In probability and statistical inference, there are two types of estimation: point and interval, the latter also being known as the confidence interval (CI). Point estimations for the parameters of a three-parameter lognormal distribution have been developed and discussed by many researchers. For example, Cohen and Whitten [11] proposed modifications of the moment and maximum likelihood estimates (MLEs). After that Cohen *et al.* [12] modified the moment estimates by replacing the function of the first-order statistic in the third moment. Singh *et al.* [2] conducted a performance evaluation of the estimates through Monte Carlo simulation. Later, interval estimations were constructed for the parameters of a three-parameter lognormal distribution. Royston [13] constructed CIs for the reference range of random samples from a three-parameter lognormal distribution. Pang *et al.* [14] used a simulation-based approach to assess the Bayesian CIs for the coefficient of variation of a three-parameter lognormal distribution as one of their proposed distributions. Next, Basak *et al.* [15] evaluated a maximum likelihood estimate created from an expectation-maximization algorithm when progressive Type-II censored samples were drawn from a three-parameter lognormal distribution while providing interval estimations for the mean, variance and threshold in a three-parameter lognormal distribution using large-sample theory. Finally, Chen and Miao [16] studied the exact CIs and exact upper CIs for the location parameter (also known as the threshold parameter) of a three-parameter lognormal distribution.

Recently, Maneerat *et al.* [17] revealed that the highest posterior density (HPD) interval-based beta prior was the best-performing method for estimating a single mean and the difference between two delta-lognormal means. After that the HPD-based normal gamma prior was developed in the comparison of the estimated rainfall dispersion between northern and northeastern regions in Thailand proposed by Maneerat *et al.* [18], while the HPD-based probability matching prior was recommended to construct the CIs for the difference between two delta-lognormal variances [19]. Unfortunately, the CI for the mean of a delta-three parameter lognormal distribution, especially when the highly skewed non-zero values are present with the zero observations, has not yet been established. Therefore, the first goal in the study was to propose a Bayesian CI (BCI) estimation-based approach (equal-tailed (ET) and HPD intervals based on different noninformative (NI) priors), generalized CI (GCI), and the method of variance estimates recovery (MOVER) for the mean of a delta-three parameter lognormal distribution. Using this as a starting point, the other goal was to estimate the weekly natural rainfall amounts during Tropical Storm Wipha in northern Thailand using our proposed methods since the observed rainfall data can be used to indicate an extreme event that can cause flooding.

The article is outlined as follows. The notation and the proposed methods in the construction of the CIs of the mean of a delta-three parameter lognormal distribution are elaborated in Section. In Sections -, details of the simulation studies, including the parameter settings

together with the criteria for assessing the proposed methods and numerical computation to identify the best-performing method, are presented. The efficacies of the proposed methods using weekly natural rainfall amounts during the period of Tropical Storm Wipha are examined in Section. Finally, this article is ended with a brief discussion and conclusions in Sections -.

## Notation and methods

Let $X = (X_1, X_2, \ldots, X_n)$ be a random sample follows a delta-three parameter lognormal distribution (DTPLN), denoted as $X \sim \text{DTPLN}(\delta, \mu_X, \sigma_X^2, a)$ where $\delta$ is the probability of having zero, $\mu_X$ is the scale parameter, $\sigma_X^2$ is the shape parameter, and $a$ is the threshold parameter or a quantity defined as a lower bound of $X$. These parameters $\delta, \mu_X, \sigma_X^2, a$ satisfy $0 < \delta < 1, \mu_X > 0$, $\sigma_X^2 > 0$ and $a \geq 0$, respectively. The distribution function of $X$ is

$$G(x; \delta, \mu_X, \sigma_X^2, a) = \begin{cases} \delta & ; x = 0 \\ 0 & ; 0 < x \leq a \\ \delta + (1-\delta)H(x; \mu_X, \sigma_X^2, a) & ; x > a \end{cases} \tag{1}$$

where $H(x; \mu_X, \sigma_X^2, a)$ is the three-parameter lognormal (TPLN) distribution, introduced by Aitchison and Brown [3], Cohen and Whitten [11] so that the probability density function of $X$ is

$$h(x; \mu_X, \sigma_X^2, a) = \frac{1}{(x-a)\sqrt{2\pi\sigma_Y^2}} \exp\left\{ -\frac{[\ln(x-a) - \mu_Y]^2}{2\sigma_Y^2} \right\} \tag{2}$$

where $\mu_Y = E[\ln(x-a)]$ and $\sigma_Y^2 = E[\ln^2(x-a)] - \{E[\ln(x-a)]\}^2$. For $x > a$, it has the relation between random variables $X$ and $Y = \ln(X-a)$, that is, $X \sim \text{TPLN}(\mu_X, \sigma_X^2, a)$ if $Y \sim N(\mu_Y, \sigma_Y^2)$. The mean of $X$ is defined as $\mu_X = (1-\delta)[a + \exp(\mu_Y + \sigma_Y^2/2)]$ which is log-transformed a

$$\theta = \ln(1-\delta) + \ln[a + \exp(\mu_Y + \sigma_Y^2/2)] \tag{3}$$

The likelihood of $X$ is

$$\begin{aligned} L(\delta, \mu_Y, \sigma_Y^2, a | \text{data}) &= \prod_{i=1}^{n} \binom{n}{x_i} \delta^{x_i}(1-\delta)^{1-x_i} \prod_{i=1}^{n_{(1)}} \frac{(2\pi\sigma_Y^2)^{-1/2}}{\exp(Y_i)} \exp\left\{ -\frac{1}{2\sigma_Y^2}[Y_i - \mu_Y]^2 \right\} \\ &= \binom{n}{n_{(0)}} \delta^{n_{(0)}}(1-\delta)^{n_{(1)}}(2\pi\sigma_Y^2)^{-n_{(1)}/2} \exp\left\{ -\frac{1}{2\sigma_Y^2} \sum_{i=1}^{n_{(1)}} [\ln(X_i - a) \right. \\ &\quad \left. -\mu_Y]^2 - \sum_{i=1}^{n_{(1)}} \ln(X_i - a) \right\} \end{aligned} \tag{4}$$

where $n_{(0)} = \{i : x_i = 0\}$ and $n_{(1)} = n - n_{(0)}$. The likelihood (4) leads to obtain the log-likelihood is

$$\begin{aligned} \ln L(\delta, \mu_Y, \sigma_Y^2, a | \text{data}) &= \text{constant.} + n_{(0)} \ln \delta + n_{(1)} \ln(1-\delta) - \frac{n_{(1)}}{2} \ln(\sigma_Y^2) \\ &\quad -\frac{1}{2\sigma_Y^2} \sum_{i=1}^{n_{(1)}} [\ln(X_i - a) - \mu_Y]^2 - \sum_{i=1}^{n_{(1)}} \ln(X_i - a) \end{aligned} \tag{5}$$

The first and second derivatives are

$$\frac{\partial}{\partial \delta} \ln L(\delta, \mu_Y, \sigma_Y^2, a | \text{data}) = \frac{n_{(0)}}{\delta} - \frac{n_{(1)}}{1 - \delta}$$

$$\frac{\partial}{\partial \mu_Y} \ln L(\delta, \mu_Y, \sigma_Y^2, a | \text{data}) = \frac{1}{\sigma_Y^2} \sum_{i=1}^{n_{(1)}} \ln(X_i - a)$$

$$\frac{\partial}{\partial \sigma_Y^2} \ln L(\delta, \mu_Y, \sigma_Y^2, a | \text{data}) = -\frac{n_{(1)}}{2\sigma_Y^2} + \frac{1}{2(\sigma_Y^2)^2} \sum_{i=1}^{n_{(1)}} [\ln(X_i - a) - \mu_Y]^2$$

$$\frac{\partial}{\partial a} \ln L(\delta, \mu_Y, \sigma_Y^2, a | \text{data}) = \sum_{i=1}^{n_{(1)}} \frac{1}{X_i - a} + \frac{1}{\sigma_Y^2} \sum_{i=1}^{n_{(1)}} \left[ \frac{\ln(X_i - a) - \mu_Y}{X_i - a} \right]$$

and

$$\frac{\partial^2}{\partial \delta^2} \ln L(\delta, \mu_Y, \sigma_Y^2, a | \text{data}) = -\frac{n_{(0)}}{\delta^2} - \frac{n}{1 - \delta}$$

$$\frac{\partial^2}{\partial \mu_Y^2} \ln L(\delta, \mu_Y, \sigma_Y^2, a | \text{data}) = -\frac{n_{(1)}}{\sigma_Y^2}$$

$$\frac{\partial^2}{\partial (\sigma_Y^2)^2} \ln L(\delta, \mu_Y, \sigma_Y^2, a | \text{data}) = \frac{n_{(1)}}{2(\sigma_Y^2)^2} - \frac{1}{(\sigma_Y^2)^3} \sum_{i=1}^{n_{(1)}} [\ln(X_i - a) - \mu_Y]^2$$

$$\frac{\partial^2}{\partial a^2} \ln L(\delta, \mu_Y, \sigma_Y^2, a | \text{data}) = \sum_{i=1}^{n_{(1)}} \frac{1}{(X_i - a)^2} + \frac{1}{\sigma_Y^2} \sum_{i=1}^{n_{(1)}} \{-1 + [\ln(X_i - a) - \mu_Y]\}$$

From $\frac{\partial}{\partial a} \ln L(\delta, \mu_Y, \sigma_Y^2, a | \text{data}) = 0$, we obtain that

$$\sum_{i=1}^{n_{(1)}} \frac{1}{X_i - a} + \frac{1}{\sigma_Y^2} \sum_{i=1}^{n_{(1)}} \left[ \frac{\ln(X_i - a) - \mu_Y}{X_i - a} \right] = 0$$

$$\sum_{i=1}^{n_{(1)}} \frac{1}{X_i - a} \left[ n_{(1)} \sum_{i=1}^{n_{(1)}} \ln(X_i - a) - \sum_{i=1}^{n_{(1)}} \ln(X_i - a) + \sum_{i=1}^{n_{(1)}} \ln^2(X_i - a) \right] \tag{6}$$

$$-n_{(1)}^{-1} \frac{\left[\sum_{i=1}^{n_{(1)}} \ln(X_i - a)\right]^2}{X_i - a} = 0$$

The estimate of threshold ($\hat{a}$) was obtained by the modified method of moments estimation, proved by Cohen and Whitten [11]. Thus, the maximum likelihood estimates (MLEs) of $\delta$, $\mu_Y$ and $\sigma_Y^2$ are

$$\hat{\delta} = \frac{n_{(0)}}{n} \tag{7}$$

$$\hat{\mu}_Y = \frac{1}{n_{(1)}} \sum_{i:x_i > 0} \ln(X_i - \hat{a}) \tag{8}$$

$$\hat{\sigma}_{Y,mle}^2 = \frac{1}{n_{(1)}} \sum_{i:x_i > 0} [\ln(X_i - \hat{a}) - \hat{\mu}_Y]^2$$

$$= \frac{1}{n_{(1)}} \sum_{i:x_i > 0} \ln^2(X_i - \hat{a}) - \left[ \frac{1}{n_{(1)}} \sum_{i:x_i > 0} \ln(X_i - \hat{a}) \right]^2 \tag{9}$$

For formulating the CIs for $\theta$, the methods are detailed as follows:

## Bayesian confidence intervals

The BCI for a parameter of interest is constructed from the posterior distribution, introduced by Gelman *et al.* [20]. Based on Bayesian approach, the $100(1 - \alpha)$% equal-tailed CI or central interval for the parameter of interest can be computed the lower and upper limits from the $100(\alpha/2)$% and $100(1 - \alpha/2)$% quantiles of the posterior probability, respectively. Box and Tiao [21] defined the HPD region (Definition 1) can be led to construct the HPD interval which is different from the equal-tailed CI.

**Definition 1**. Let $p(\theta|y)$ be a posterior density function. A region $R$ in the parameter space of $\theta$ is called a HPD region of content $(1 - \alpha)$ if

*i)* $Pr(\theta \in R|y) = 1 - \alpha$,

*ii)* For $\theta_1 \in R$ and $\theta_2 \notin R$, $p(\theta_1|y) \geq p(\theta_2|y)$.

The HPD region is defined as the value set that contains the $100(1 - \alpha)$% of the posterior probability, importantly its density within the region is never less than that outside. In the present study, the equal-tailed CIs and HPD intervals for the log-transformed mean of a delta-three parameter lognormal distribution are proposed and constructed using the noninformative (NI) priors as follows:

**NI1 prior.** The NI1 prior is derived from the square root of the Fisher information matrix of $(\mu_Y, \sigma_Y^2, \delta)$, given by

$$P_{(NI1)}(\theta) \propto \sqrt{I(\mu_Y)I(\sigma_Y^2)I(\delta)} = \sqrt{\sigma_Y^{-3}\delta^{-1}(1 - \delta)} \tag{10}$$

Recall that the likelihood is given by

$$
\begin{aligned}
L(\delta, \mu_Y, \sigma_Y^2, a|\text{data}) \quad \propto \quad & \delta^{n_{(0)}}(1 - \delta)^{n_{(1)}}(\sigma_Y^2)^{-n_{(1)}/2}\exp\left\{-\frac{1}{2\sigma_Y^2}\sum_{i=1}^{n_{(1)}}[\ln(X_i - a) - \hat{\mu}_Y]^2 - \sum_{i=1}^{n_{(1)}}\ln(X_i - a)\right\} \\
\propto \quad & \delta^{n_{(0)}}(1 - \delta)^{n_{(1)}}(\sigma_Y^2)^{-n_{(1)}/2}\exp\left\{-\frac{1}{2\sigma_Y^2}\left[(n_{(1)} - 1)\hat{\sigma}_Y^2\right.\right. \\
& \left.\left. +n_{(1)}(\hat{\mu}_Y - \mu_Y)^2\right] - \sum_{i=1}^{n_{(1)}}\ln(X_i - a)\right\}
\end{aligned}
\tag{11}
$$

where $(n_{(1)} - 1)\hat{\sigma}_Y^2 = \sum_{i=1}^{n_{(1)}}[\ln(X_i - a) - \hat{\mu}_Y]^2$. The likelihood (11) is updated with the NI1 prior (10) to obtain the posterior of $(\mu_Y, \sigma_Y^2, \delta, a)$, that is

$$
\begin{aligned}
P_{(NI1)}(\delta, \mu_Y, \sigma_Y^2, a|\text{data}) \quad \propto \quad & \delta^{n_{(0)}-\frac{1}{2}}(1 - \delta)^{n_{(1)}+\frac{1}{2}}(\sigma_Y^2)^{-\frac{n_{(1)}+3}{2}}\exp\left\{-\frac{1}{2\sigma_Y^2}\left[(n_{(1)} - 1)\hat{\sigma}_Y^2\right.\right. \\
& \left.\left. +n_{(1)}(\hat{\mu}_Y - \mu_Y)^2\right] - \sum_{i=1}^{n_{(1)}}\ln(X_i - a)\right\}
\end{aligned}
\tag{12}
$$

The posterior of $\delta$ is

$$P(\delta|\text{data}) \propto \delta^{\left(n_{(0)}+\frac{1}{2}\right)-1}(1 - \delta)^{\left(n_{(1)}+\frac{3}{2}\right)-1} \tag{13}$$

It can be implied that $\delta^{post1} \sim \text{beta}\left(n_{(0)} + \frac{1}{2}, n_{(1)} + \frac{3}{2}\right)$. From Eq (12), the posterior of $a|\text{data}$ becomes

$$
\begin{aligned}
P_{(NI1)}(a|\text{data}) \quad &\propto \quad \int \int P_{(NI1)}(\delta, \mu_Y, \sigma_Y^2, a|\text{data}) \mathrm{d}\mu_Y \mathrm{d}\sigma_Y^2 \\[2mm]
&\propto \quad \int (\sigma_Y^2)^{-\frac{n_{(1)}+3}{2}} \exp\left\{ -\frac{(n_{(1)}-1)\hat{\sigma}_Y^2}{2\sigma_Y^2} - \sum_{i=1}^{n_{(1)}} \ln(X_i - a) \right\} \\[2mm]
&\qquad \int -\frac{n_{(1)}}{2\sigma_Y^2}(\hat{\mu}_Y - \mu_Y)^2 \mathrm{d}\mu_Y \mathrm{d}\sigma_Y^2 \\[2mm]
&\propto \quad \int (\sigma_Y^2)^{-\frac{n_{(1)}+3}{2}} \exp\left\{ -\frac{(n_{(1)}-1)\hat{\sigma}_Y^2}{2\sigma_Y^2} - \sum_{i=1}^{n_{(1)}} \ln(X_i - a) \right\} \left( \frac{2\pi\sigma_Y^2}{n_{(1)}} \right)^{1/2} \mathrm{d}\sigma_Y^2 \qquad (14) \\[2mm]
&\propto \quad \exp\left\{ -\sum_{i=1}^{n_{(1)}} \ln(X_i - a) \right\} \int (\sigma_Y^2)^{-\frac{n_{(1)}}{2}-1} \exp\left\{ -\frac{(n_{(1)}-1)\hat{\sigma}_Y^2}{2\sigma_Y^2} \right\} \mathrm{d}\sigma_Y^2 \\[2mm]
&\propto \quad \exp\left\{ -\sum_{i=1}^{n_{(1)}} \ln(X_i - a) \right\}
\end{aligned}
$$

where $a^{post1} \propto \exp\left\{ -\sum_{i=1}^{n_{(1)}} \ln(X_i - a) \right\}$ which can be obtained its random samples using Metropolis algorithm. Let $A = (n_{(1)} - 1)\hat{\sigma}_Y^2 + n_{(1)}(\hat{\mu}_Y - \mu_Y)^2$, $Z = \frac{A}{2\sigma_Y^2}$ such that $\frac{\mathrm{d}Z}{\mathrm{d}\sigma_Y^2} = \frac{2Z^2}{A}$ leads to obtain $\mathrm{d}\sigma_Y^2 = \frac{A}{2Z^2}\mathrm{d}Z$. The posterior of $\mu_Y$ is

$$
\begin{aligned}
P_{(NI1)}(\mu_Y|\text{data}) \quad &\propto \quad \int \int P_{(NI1)}(\delta, \mu_Y, \sigma_Y^2, a|\text{data}) \mathrm{d}a \mathrm{d}\sigma_Y^2 \\[2mm]
&\propto \quad \int (\sigma_Y^2)^{-\frac{n_{(1)}+3}{2}} \exp\left\{ -\frac{A}{2\sigma_Y^2} \right\} \int \exp\left\{ -\sum_{i=1}^{n_{(1)}} \ln(X_i - a) \right\} \mathrm{d}a \mathrm{d}\sigma_Y^2 \\[2mm]
&\propto \quad \int (\sigma_Y^2)^{-\frac{n_{(1)}+3}{2}} \exp\left\{ -\frac{A}{2\sigma_Y^2} \right\} \frac{\exp\left\{ -\sum_{i=1}^{n_{(1)}} \ln(X_i - a) \right\}}{\sum_{i=1}^{n_{(1)}} (X_i - a)^{-1}} \mathrm{d}\sigma_Y^2 \\[2mm]
&\propto \quad \int (\sigma_Y^2)^{-\frac{n_{(1)}+3}{2}} \exp\left\{ -\frac{A}{2\sigma_Y^2} \right\} \mathrm{d}\sigma_Y^2 \\[2mm]
&\propto \quad \int A^{-\frac{n_{(1)}+3}{2}} \exp\{-Z\} \frac{A}{2Z^2} \mathrm{d}Z \qquad (15) \\[2mm]
&\propto \quad A^{-\frac{n_{(1)}+1}{2}} \int Z^{(n_{(1)}+\frac{1}{2})-1} \exp\{-Z\} \mathrm{d}Z \\[2mm]
&\propto \quad [(n_{(1)} - 1)\hat{\sigma}_Y^2 + n_{(1)}(\hat{\mu}_Y - \mu_Y)^2]^{-\frac{n_{(1)}+1}{2}} \\[2mm]
&\propto \quad [(n_{(1)} - 1)\hat{\sigma}_Y^2]^{-\frac{n_{(1)}+1}{2}} \left[ 1 + \frac{n_{(1)}(\hat{\mu}_Y - \mu_Y)^2}{(n_{(1)} - 1)\hat{\sigma}_Y^2} \right]^{-\frac{n_{(1)}+1}{2}} \\[2mm]
&\propto \quad \left[ 1 + \frac{n_{(1)}(\hat{\mu}_Y - \mu_Y)^2}{(n_{(1)} - 1)\hat{\sigma}_Y^2} \right]^{-\frac{n_{(1)}+1}{2}}
\end{aligned}
$$

Thus, $\mu_Y^{post} \sim t_{df}(\hat{\mu}_Y, \hat{\sigma}_Y^2/n_{(1)})$; $df = n_{(1)} - 1$. The posterior of $\sigma_Y^2 |$data is

$$
\begin{aligned}
P_{(NI1)}(\sigma_Y^2|\text{data}) &\propto \int\int P_{(NI1)}(\delta, \mu_Y, \sigma_Y^2, a|\text{data})\mathrm{d}\mu_Y\mathrm{d}a \\
&\propto \int (\sigma_Y^2)^{-\frac{n_{(1)}+3}{2}}\exp\left\{-\frac{(n_{(1)}-1)\hat{\sigma}_Y^2}{2\sigma_Y^2} - \sum_{i=1}^{n_{(1)}}\ln(X_i-a)\right\} \\
&\quad\int -\frac{n_{(1)}}{2\sigma_Y^2}(\hat{\mu}_Y-\mu_Y)^2\mathrm{d}\mu_Y\mathrm{d}a \\
&\propto (\sigma_Y^2)^{-\frac{n_{(1)}}{2}-1}\exp\left\{-\frac{(n_{(1)}-1)\hat{\sigma}_Y^2}{2\sigma_Y^2}\right\}\int\exp\left\{-\sum_{i=1}^{n_{(1)}}\ln(X_i-a)\right\}\mathrm{d}a \\
&\propto (\sigma_Y^2)^{-\frac{n_{(1)}}{2}-1}\exp\left\{-\frac{(n_{(1)}-1)\hat{\sigma}_Y^2}{2\sigma_Y^2}\right\}
\end{aligned}
\tag{16}
$$

where $\sigma_Y^{2(post)} \sim IG(\alpha, \beta)$; $\alpha = n_{(1)}/2$ and $\beta = [n_{(1)} - 1]\hat{\sigma}_Y^2/2$. The posterior of $\theta$ based on NI1 can be expressed as

$$
\theta^{post1} = \ln(\delta^{post1}) + \ln[a^{post1} + \exp(\mu_Y^{post1} + \sigma_Y^{2(post1)}/2)]
\tag{17}
$$

Finally, the $100(1 - \alpha)$%BCIs for $\theta$ based on NI1 prior are

$$
\text{BCI}_{\text{ET-NI1}} = [\theta_{\alpha/2}^{post1}, \theta_{1-\alpha/2}^{post1}]
\tag{18}
$$

$$
\text{BCI}_{\text{HPD-NI1}} = [\theta_l^{post1}, \theta_u^{post1}]
\tag{19}
$$

where $\theta_\alpha^{post1}$ denotes the $\alpha^{\text{th}}$ quantile of $\theta^{post1}$ and $Pr(\theta_l^{post1} < \theta < \theta_u^{post1}) = 1 - \alpha$.

**NI2 prior.** This prior belief is obtained from the $\delta$, $a$, $\mu_Y$ and $\sigma_Y^2$ are treated as random variables of the beta, uniform, normal and gamma distributions, denoted as beta$(c, c)$, $U(a' = 0, b' = 1)$, $N(\mu_Y, 1/k\eta)$ and $G(a, b)$, respectively, where $c = 1/3 + w_{\alpha/2}^2/6$ and $\eta = 1/\sigma_Y^2$. The prior of $\delta$ was derived by Jin *et al.* [22]. The prior of $\theta = (\delta, \mu_Y, \sigma_Y^2)$ is

$$
\begin{aligned}
P(\delta, a, \mu_Y, \eta) &= P(\delta; c, c)P(a; a', b')P(\mu_Y|\eta; \mu_{Y,0}, k_0\eta)P(\eta; a_0, b_0) \\
&= \left\{\frac{\Gamma(2c)}{\Gamma^2(c)}\delta^{c-1}(1-\delta)^{c-1}\right\}\left\{\frac{1}{b'-a'}\right\}\left\{\frac{1}{\sqrt{2\pi/(k_0\eta)}}\exp\left[-\frac{1}{2/(k_0\eta)}\right.\right. \\
&\quad \left.\left.(\mu_Y-\mu_{Y,0})^2\right]\right\}\left\{\frac{b_0}{\Gamma(a_0)}\eta^{a_0-1}\exp[-b_0\eta]\right\} \\
&\propto \delta^{c-1}(1-\delta)^{c-1}\eta^{a_0-1/2}\exp\left[-\frac{\eta}{2}\left(k_0(\mu_Y-\mu_{Y,0})^2 + 2b_0\right)\right]
\end{aligned}
\tag{20}
$$

When $k_0 = 0$, $a_0 = -1/2$ and $b_0 = 0$, the NI2 prior of $(\mu_Y, \sigma^2, \delta)$ is derived from $(\mu_Y, \sigma_Y^2) \sim NG(\mu_Y, \eta|\mu_Y, k_0 = 0, a_0 = -1/2, b_0 = 0)$, and $\delta \sim$ beta$(c, c)$ as

$$
P_{(NI2)}(\theta) \propto \delta^{c-1}(1-\delta)^{c-1}\eta^{-1}
\tag{21}
$$

which is combined with the likelihood (11) in term of $\eta$ as

$$
\begin{aligned}
L(\delta, \mu_Y, \eta, a | \text{data}) \quad \propto \quad & \delta^{n_{(0)}} (1-\delta)^{n_{(1)}} \eta^{\frac{n_{(1)}}{2}} \exp\left[-\frac{\eta}{2}\left\{(n_{(1)}-1)\hat{\sigma}_Y^2 + n_{(1)}(\hat{\mu}_Y - \mu_Y)^2\right\}\right. \\
& \left. -\sum_{i=1}^{n_{(1)}} \ln(X_i - a)\right]
\end{aligned}
\tag{22}
$$

Meanwhile, the posterior of $\theta$ is

$$
\begin{aligned}
P_{(NI2)}(\delta, \mu_Y, \eta, a | \text{data}) \quad \propto \quad & \delta^{n_{(0)}+c-1}(1-\delta)^{n_{(1)}+c-1} \eta^{\frac{n_{(1)}}{2}-1} \exp\left[-\frac{\eta}{2}\left\{(n_{(1)}-1)\hat{\sigma}_Y^2\right.\right. \\
& \left.\left. +n_{(1)}(\hat{\mu}_Y - \mu_Y)^2\right\} - \sum_{i=1}^{n_{(1)}} \ln(X_i - a)\right] \\
\propto \quad & \delta^{n_{(0)}+c-1}(1-\delta)^{n_{(1)}+c-1} \eta^{\frac{n_{(1)}-1}{2}-1} \exp\left[-\frac{\eta(n_{(1)}-1)\hat{\sigma}_Y^2}{2}\right] \\
& \sqrt{\eta}\exp\left[-\frac{\eta n_{(1)}}{2}(\hat{\mu}_Y - \mu_Y)^2\right]\exp\left[-\sum_{i=1}^{n_{(1)}}\ln(X_i - a)\right]
\end{aligned}
\tag{23}
$$

This leads to obtain the posterior of $\delta$ is

$$
P(\delta | \text{data}) \propto \delta^{(n_{(0)}+c)-1}(1-\delta)^{(n_{(1)}+c)-1}
\tag{24}
$$

This is $\delta^{post2} \sim \text{beta}(n_{(0)}+ c, n_{(1)}+ c)$. The posterior of $a|$data becomes

$$
\begin{aligned}
P_{(NI2)}(a | \text{data}) \quad \propto \quad & \int P_{(NI2)}(\eta, a | \text{data})\,d\eta \\
\propto \quad & \int \eta^{\frac{n_{(1)}-1}{2}-1}\exp\left[-\frac{\eta(n_{(1)}-1)\hat{\sigma}_Y^2}{2}\right]\exp\left[-\sum_{i=1}^{n_{(1)}}\ln(X_i - a)\right]d\eta \\
\propto \quad & \exp\left[-\sum_{i=1}^{n_{(1)}}\ln(X_i - a)\right]
\end{aligned}
\tag{25}
$$

which can be written as $a^{post2} \propto \exp\left\{-\sum_{i=1}^{n_{(1)}} \ln(X_i - a)\right\}$. Let $W = \beta + [n_{(1)}(\hat{\mu}_Y - \mu_Y)^2/2]$ and $T = \eta W$ such that $d\eta = \frac{1}{W}dT$; $\beta = (n_{(1)}-1)\hat{\sigma}_Y^2/2 = \sum_{i=1}^{n_{(1)}}[\ln(X_i - a) - \hat{\mu}_Y]^2/2$. The

posterior of $\mu|$data is

$$
\begin{aligned}
P_{(NI2)}(\mu|\text{data}) &\propto \int\int P_{(NI2)}(\delta,\mu_Y,\eta,a|\text{data})\mathrm{d}a\mathrm{d}\eta \\
&\propto \int \eta^{\frac{n_{(1)}}{2}-1}\exp[-\eta W]\int \exp\left[-\sum_{i=1}^{n_{(1)}}\ln(X_i-a)\right]\mathrm{d}a\mathrm{d}\eta \\
&\propto \int \left(\frac{T}{W}\right)^{\frac{n_{(1)}}{2}-1}\frac{1}{W}\exp[-T]\mathrm{d}T \\
&\propto \left(\frac{1}{W}\right)^{\frac{n_{(1)}}{2}}\int T^{\frac{n_{(1)}}{2}-1}\exp[-T]\mathrm{d}T \\
&\propto \left[\beta+\frac{n_{(1)}}{2}(\hat{\mu}_Y-\mu_Y)^2\right]^{-\frac{n_{(1)}-1}{2}+\frac{1}{2}} \\
&\propto \left[1+\frac{1}{2(n_{(1)}-1)}\frac{n_{(1)}(n_{(1)}-1)}{\beta}(\hat{\mu}_Y-\mu_Y)^2\right]^{-\frac{n_{(1)}-1}{2}+\frac{1}{2}}
\end{aligned}
$$

which is the Student' t distribution with $n_{(1)}-1$ the degrees of freedom (df), denoted as $\mu_Y^{post2}\sim t_{n_{(1)}-1}(\hat{\mu}_Y,\beta/[n_{(1)}(n_{(1)}-1)])$. From Eq (23), the posterior of $\eta|$data is

$$
\begin{aligned}
P_{(NI2)}(\eta|\text{data}) &\propto \int\int P_{(NI2)}(\delta,\mu_Y,\eta,a|\text{data})\mathrm{d}\mu\mathrm{d}a \\
&\propto \int\int \eta^{\frac{n_{(1)}-1}{2}-\frac{1}{2}}\exp\left[-\frac{\eta(n_{(1)}-1)\hat{\sigma}_Y^2}{2}\right]\exp\left[-\sum_{i=1}^{n_{(1)}}\ln(X_i-a)\right] \\
&\quad \exp\left[-\frac{\eta n_{(1)}}{2}(\hat{\mu}_Y-\mu_Y)^2\right]\mathrm{d}\mu\mathrm{d}a \\
&\propto \int \eta^{\frac{n_{(1)}-1}{2}-1}\exp\left[-\frac{\eta(n_{(1)}-1)\hat{\sigma}_Y^2}{2}\right]\exp\left[-\sum_{i=1}^{n_{(1)}}\ln(X_i-a)\right]\mathrm{d}a \\
&\propto \eta^{\frac{n_{(1)}-1}{2}-1}\exp\left[-\frac{\eta(n_{(1)}-1)\hat{\sigma}_Y^2}{2}\right]\frac{\exp[-\sum_{i=1}^{n_{(1)}}\ln(X_i-a)]}{\sum_{i=1}^{n_{(1)}}(X_i-a)^{-1}} \\
&\propto \eta^{\frac{n_{(1)}-1}{2}-1}\exp\left[-\frac{\eta(n_{(1)}-1)\hat{\sigma}_Y^2}{2}\right]
\end{aligned}
\tag{26}
$$

It can be concluded that $\eta^{post2}\sim G\left(\frac{n_{(1)}-1}{2},\frac{(n_{(1)}-1)\hat{\sigma}_Y^2}{2}\right)$. Thus, the posterior of $\sigma_Y^2$ is

$$
\sigma_Y^{2(post2)}\sim IG\left(\frac{n_{(1)}-1}{2},\frac{(n_{(1)}-1)\hat{\sigma}_Y^2}{2}\right)
\tag{27}
$$

The posterior of $\theta$ based on NI2 can be written as

$$
\theta^{post2}=\ln(\delta^{post2})+\ln[a^{post2}+\exp(\mu_Y^{post2}+\sigma_Y^{2(post2)}/2)]
\tag{28}
$$

Hence, the $100(1-\alpha)\%$BCIs for $\theta$ based on NI2 prior are

$$
\text{BCI}_{\text{ET}-\text{NI2}} = [\theta_{\alpha/2}^{post2},\theta_{1-\alpha/2}^{post2}]
\tag{29}
$$

$$
\text{BCI}_{\text{HPD}-\text{NI2}} = [\theta_l^{post2},\theta_u^{post2}]
\tag{30}
$$

where $\theta_{\alpha}^{post2}$ denotes the $\alpha^{\text{th}}$ quantile of $\theta^{post2}$ and $P(\theta_l^{post2} < \theta < \theta_u^{post2}) = 1 - \alpha$. Algorithm 1 describes the steps for constructing BCIs for the delta-three parameter lognormal means.

**Algorithm 1**: BCIs

```
1: Compute the unbiased estimates δ̂, μ̂_Y, σ̂²_Y, and â.
2: For NI1 prior, generate the posterior distributions of δ, a, μ_Y, and
   σ²_Y, denoted by δ^post1, a^post1 μ_Y^post1, and σ_Y^2(post1) in Eqs (13), (14), (15)
   and (16), respectively.
3: For NI2 prior, generate the posterior distributions of δ, a, μ_Y, and
   σ²_Y, denoted by δ^post2, a^post2 μ_Y^post2, and σ_Y^2(post2) in Eqs (24), (25), (26)
   and (27), respectively.
4: Compute θ^post1 and θ^post2 based on NI1 and NI2 priors, respectively.
5: Repeat 2-4 a number of times, say, m.
6: For m times, compute the 100(1 - α)% ET and HPD intervals for θ
   based on priors: BCI_ET-NI1, BCI_HPD-NI1, BCI_ET-NI2 and BCI_HPD-NI2.
```

## Generalized confidence interval

The GCI is established based on the concept of generalized pivotal quantity (GPQ), defined by Weerahandi [23]. The CI of $\theta$ can be constructed using GCI. Recall that $X \sim \text{TPLN}(\mu_X, \sigma_X^2, a)$ if $Y = \ln(X - a)$ be a random variable of normal distribution with mean $\mu_Y$ and variance $\sigma_Y^2$. Cohen [24], Cohen *et al.* [12] and Cohen and Whitten [11] derived the MLE of threshold (6), and the asymptotic variance of $\hat{a}$ is based on the Fisher information matrix, given by

$$\sigma_{\hat{a}}^2 = \frac{\sigma_Y^2 \exp(2\mu_Y - \sigma_Y^2)}{n_{(1)}[\exp(\sigma_Y^2)(1 + \sigma_Y^2) - 2\sigma_Y^2 - 1]} \tag{31}$$

By replacing the estimates $\hat{\mu}_Y$ and $\hat{\sigma}_Y^2$, the $\sigma_{\hat{a}}^2$ is estimated and denoted by $\hat{\sigma}_{\hat{a}}^2$. Let $\hat{a}$ be a random variable. From the approximation result, it is transformed as

$$T = \frac{\hat{a} - a}{\sqrt{\hat{\sigma}_{\hat{a}}^2}} = \frac{(\hat{a} - a)/\sqrt{\sigma_{\hat{a}}^2}}{\sqrt{\hat{\sigma}_{\hat{a}}^2/\sigma_{\hat{a}}^2}} = \frac{Z}{\sqrt{V/(n_{(1)} - 2)}} \tag{32}$$

which has a Student's t distribution with $n_{(1)} - 2$ df, where $Z = (\hat{a} - a)/\sqrt{\hat{\sigma}_{\hat{a}}^2}$ and $V = (n_{(1)} - 2)\hat{\sigma}_{\hat{a}}^2/\sigma_{\hat{a}}^2$ are independent random variables of standard normal and chi-square distribution with $n_{(1)} - 2$ df, denoted by $Z \sim N(0, 1)$ and $V \sim \chi_{n_{(1)}-2}^2$, respectively. By the information of pivotal quantity $T$, the GPQ of $a$ is

$$R_a = \hat{a} - T\sqrt{\hat{\sigma}_{\hat{a}}^2} \tag{33}$$

Furthermore, Wu and Hsieh [25] proposed the GPQs of $\delta$ and $(\mu_Y, \sigma_Y^2)$, defined as

$$R_\delta = \sin^2\left[\arcsin\sqrt{\hat{\delta}} - \frac{K}{2\sqrt{n_{(1)}}}\right] \tag{34}$$

$$R_{\mu_Y} = \hat{\mu}_Y - W\sqrt{R_{\sigma_Y^2}/n_{(1)}} \tag{35}$$

$$R_{\sigma_Y^2} = (n_{(1)} - 1)\hat{\sigma}_Y^2/U \tag{36}$$

where $K = 2\sqrt{n_{(1)}}(\arcsin\sqrt{\hat{\delta}} - \arcsin\sqrt{\delta}) \sim N(0, 1)$, $W = (\hat{\mu}_Y - \mu_Y)/\sqrt{R_{\sigma_Y^2}/n_{(1)}} \sim N(0, 1)$

and $U \sim \chi^2_{n_{(1)}-1}$. The GPQ of $\theta$ is

$$R_\theta = \ln(1 - R_\delta) + \ln[R_a + \exp(R_{\mu_Y} + R_{\sigma_Y^2}/2)] \tag{37}$$

which satisfies the conditions of Weerahandi [23], i.e., the distribution of $R_\theta$ is free from all unknown parameters, and the observed value of $R_\theta$ depends only on the parameters of interest. Therefore, the $100(1 - \alpha)$% GCI for $\theta$ is given by

$$\text{GCI} = [R_\theta(\alpha/2), R_\theta(1 - \alpha/2)] \tag{38}$$

where $R_\theta(\alpha)$ denotes the $\alpha^{\text{th}}$ quantile of $R_\theta$. The steps for computing GCI for $\theta$ are detailed in Algorithm 2.

**Algorithm 2**: GCI
```
1: Generate T ∼ t_n1-2, K, W ∼ N(0, 1) and U ∼ χ²_n(1)-1.
2: Compute the GPQs of a, δ, μ_Y and σ²_Y, denoted as R_a, R_δ, R_μ_Y, and R_σ²_Y,
respectively.
3: Compute the GPQ of θ, denoted as R_θ.
4: Repeat 1-3 a number of times, say, m.
5: For m times, compute the 100(1 − α)%GCI for θ in Eq (38).
```

## Method of variance estimates recovery

Let $\lambda_i$ be the parameter of interest for the population $i$; $i = 1, 2, \ldots, p$. Also, let $\hat{\lambda}_i$ be the point estimate of $\lambda_i$. The MOVER interval for the function of parameters $\lambda_i$ is a closed-form CI constructed by obtaining the variance estimates $\widehat{Var}(\hat{\lambda}_i)$ at the neighborhood of the lower and upper limits separately (to recover from confidence limits), given in Zou and Donner [26] and Zou *et al.* [27]. Thus, $100(1 - \alpha)$% MOVER interval for $\lambda$ is

$$\begin{aligned}
\text{MOVER} \quad = \quad & [(\hat{\lambda}_1 + \hat{\lambda}_2) - \sqrt{(\hat{\lambda}_1 - l_{\lambda_1})^2 + (\hat{\lambda}_2 - l_{\lambda_2})^2}, \\
& (\hat{\lambda}_1 + \hat{\lambda}_2) + \sqrt{(u_{\lambda_1} - \hat{\lambda}_1)^2 + (u_{\lambda_2} - \hat{\lambda}_2)^2}]
\end{aligned} \tag{39}$$

where $[l_{\lambda_i}, u_{\lambda_i}]$ be the $100(1 - \alpha)$% CIs for $\lambda_i$. Approximate closed-form CI for the logarithm of delta-three parameter lognormal mean is considered and developed using the MOVER. Recall that

$$\theta = \ln \delta' + \ln[a + \exp(\mu_Y + \sigma_Y^2/2)] = \ln \theta_1 + \ln \theta_2 \tag{40}$$

where $\delta' = 1 - \delta$. Let $\hat{\theta} = \ln \hat{\theta}_1 + \ln \hat{\theta}_2$ be the point estimate of $\theta$; $\hat{\theta}_1 = \hat{\delta}'$ and $\hat{\theta}_2 = \hat{a} + \exp(\hat{\mu}_Y + \hat{\sigma}_Y^2/2)$. The MOVER intervals for $\ln \theta_1$ and $\ln \theta_2$ are described as follows. First, the $100(1 - \alpha)$% Wilson interval for $\ln \theta_1$ is proposed by Zou *et al.* [27], given by

$$\begin{aligned}
\text{CI}_{\ln \theta_1} \quad = \quad & [l_{\ln \theta_1}, u_{\ln \theta_1}] \\
= \quad & \ln\left[(n_{(1)} + k_{\alpha/2}^2/2) \pm k_{\alpha/2}\sqrt{\frac{n_{(0)}n_{(1)}}{n} + \frac{k_{\alpha/2}^2}{4}}\right]/(n + k_{\alpha/2}^2)
\end{aligned} \tag{41}$$

where $k_\alpha$ denotes the $\alpha^{\text{th}}$ quantile of standard normal distribution. Next, the $100(1 - \alpha)$%

MOVER interval for $\ln\theta_2$

$$
\begin{aligned}
\mathrm{CI}_{\ln\theta_2} &= [l_{\ln\theta_2}, u_{\ln\theta_2}] \\
&= \ln\left[\hat{\theta}_2 - \sqrt{(\hat{a}-l_a)^2 + (\mu_Y + \sigma_Y^2/2 - l_{\mu_Y+\sigma_Y^2/2})^2},\right. \\
&\qquad \left.\hat{\theta}_2 + \sqrt{(u_a-\hat{a})^2 + (u_{\mu_Y+\sigma_Y^2/2} - \mu_Y - \sigma_Y^2/2))^2}\right]
\end{aligned}
\tag{42}
$$

where

$$
[l_a, u_a] = [\hat{a} - t_{\alpha/2,df}\sqrt{\hat{\sigma}_{\hat{a}}^2}, \hat{a} - t_{1-\alpha/2,df}\sqrt{\hat{\sigma}_{\hat{a}}^2}]
\tag{43}
$$

$$
\begin{aligned}
[l_{\mu_Y+\sigma_Y^2/2}, u_{\mu_Y+\sigma_Y^2/2}] &= \left[\hat{\mu}_Y + \hat{\sigma}_Y^2/2 - \sqrt{\frac{w_{\alpha/2}^2\hat{\sigma}_Y^2}{n_{(1)}} + \frac{\hat{\sigma}_Y^4}{2}\left(1 - \frac{n_{(1)}-1}{\chi_{1-\alpha/2,n_{(1)}-1}^2}\right)^2},\right. \\
&\qquad \left.\hat{\mu}_Y + \hat{\sigma}_Y^2/2 + \sqrt{\frac{w_{\alpha/2}^2\hat{\sigma}_Y^2}{n_{(1)}} + \frac{\hat{\sigma}_Y^4}{2}\left(1 - \frac{n_{(1)}-1}{\chi_{\alpha/2,n_{(1)}-1}^2} - 1\right)^2}\right]
\end{aligned}
\tag{44}
$$

Note that $t_{\alpha,df}$ and $w_\alpha$ denote the $\alpha^{\text{th}}$ quantile of Student's t distribution with $n_{(1)} - 2$ df and standard normal distributions, respectively. The $[l_{\mu_Y+\sigma_Y^2/2}, u_{\mu_Y+\sigma_Y^2/2}]$ is given in Zou *et al.* [27]. Applying Eq (39), the $100(1-\alpha)\%$ MOVER interval for $\theta$ is

$$
\begin{aligned}
\mathrm{MOVER} &= [\ln\hat{\theta}_1 + \ln\hat{\theta}_2 - \sqrt{(\ln\hat{\theta}_1 - l_{\ln\theta_1})^2 + (\ln\hat{\theta}_2 - l_{\ln\theta_2})^2},\\
&\qquad \ln\hat{\theta}_1 + \ln\hat{\theta}_2 + \sqrt{(u_{\ln\theta_1} - \ln\hat{\theta}_1)^2 + (u_{\ln\theta_2} - \ln\hat{\theta}_2)^2}]
\end{aligned}
\tag{45}
$$

The MOVER for $\theta$ can be computed in Algorithm 3.

**Algorithm 3**: MOVER
1) Compute the CIs for $a$ and $\mu_Y + \sigma_Y^2/2$ in Eqs (43) and (44), respectively.
2) Compute $\mathrm{CI}_{\ln\theta_1}$ and $\mathrm{CI}_{\ln\theta_2}$.
3) Compute the $100(1-\alpha)\%$MOVER for $\theta$ in Eq (45).

## Simulation studies

Simulation studies were conducted to calculate the performances of the methods: the coverage probabilities (CPs) and expected lengths (ELs) of BCIs (HPD and ET intervals)-based NI1 and NI2 priors, GCI, and MOVER for the logarithm of the delta-three parameter lognormal mean. Both performances are defined as follows:

CP: the proportion of intervals in which the true parameter falls within the intervals.

EL: the average lengths of simulated intervals.

Monte Carlo simulation studies were undertaken to compare the performances of our proposed methods and provide insight into their sampling behavior. In the comparison of the methods, a CI with a CP close to the nominal level 0.95 and the narrowest EL are the criteria for the best performance. In the simulation studies, the values of the threshold parameter were chosen as $a = 1, 5, 15$. For each threshold value, the parameter combinations were sample sizes $n = 30, 50, 100$; proportion of zero $\delta = 10\%, 30\%, 50\%$; mean $\mu = 2$ and variance $\sigma^2 = 0.3, 0.5, 0.8, 1.0, 2.0$. For each set of parameter settings, 5,000 simulation runs were generated and

5,000 GPQs were fixed for the GCI. The steps of the simulation study were executed as shown in Algorithm 4.

**Algorithm 4**
```
1: Generate X ~ DTPLN(δ, μ_X, σ²_X, a).
2: Compute the unbiased estimates δ̂, μ̂_Y, σ̂²_Y, and â.
3: Compute the CIs: BCIs, GCI and MOVER in Algorithms 1, 2 and 3,
   respectively.
4: For the 5000 generated values, the CIs in Step (3) are computed.
5: Computed the estimated CPs and ELs of the CIs in Step (4).
```

## Monte Carlo simulation results

The simulation results for threshold $a = 1$ (Table 1 and Fig 1) show that GCI and MOVER generated CPs close to nominal level 0.95 when the variance was small for all of the proportions of zero observations, although those of the BCIs (HPD and ET intervals based on the NI1 and NI2 priors) were under it. For threshold $a = 5$ (Table 2 and Fig 2),the CP and EL performances of HPD-NI1, ET-NI1, GCI, and MOVER were better than $a = 1$, while HPD-NI1 performed the best in terms of EL for small-to-large sample sizes except for large variance. For a large threshold $a = 15$ (Table 3 and Fig 3), HPD-NI1 performed better than the other methods with a CP close to the nominal level and the narrowest EL when the variance was small-to-medium.

## An illustrative example

We applied the CIs constructed with the proposed methods to real-world data. In the week 29 July to 4 August 2019, Tropical Storm Wipha moved from Vietnam to northern Thailand, thereby putting the area at high risk of flash floods and landslides caused by heavy rain [1]. Thus, predicting the weekly natural rainfall data in the above-mentioned period is of interest. Data on the weekly rainfall during this period was collected by the Thailand Meteorological Department (TMD) (Table 4). The northern station includes 62 substations: 55 with positive rainfall records (88.71%) and the rest with no recorded rainfall.

By applying the theory in Section and with known $\hat{a}$, the weekly positive rainfall data follow a normal distribution when they are log-transformed as $\ln(X - \hat{a})$. It is possible that this positive rainfall data have a lognormal distribution (the histogram and the empirical cumulative distribution function (CDF) plots in Fig 4). To determine which model fits the positive rainfall data, Nguyen [28] suggested that it might be insufficient to use the probability value (p-value) for decision-making alone in statistical testing of hypotheses. Thus, the Akaike information criterion (AIC) and Bayesian information criterion (BIC) are used to avoid using the p-value for model evaluation. Akaike [29] and Stone [30] defined the AIC and BIC which are the methods for scoring and selecting a suitable model derived from frequentist and Bayesian probabilities, respectively. Let $\theta$ be the set (vector) of model parameters and $L(\hat{\theta})$ be the likelihood of the candidate model when evaluated at the MLE of $\theta$. The AIC and BIC of a model are expressed as

$$\text{AIC} = -2\ln L(\hat{\theta}) + 2k \tag{46}$$

$$\text{BIC} = -2\ln L(\hat{\theta}) + 2k\ln(n) \tag{47}$$

where $k$ stands for the number of estimated parameters in the candidate model and $n$ stands for the number of recorded measurements. The AIC and BIC results, in Table 5, reveal that the reduced rainfall data $(X - \hat{a})$ fit a lognormal distribution. When factoring in the empty rainfall records, the weekly positive rainfall data in the week 29 July to 4 August 2019 follow a

**Table 1. CP and EL performances of 95% CI for θ: a = 1.**

| a = 1 | | | CP | | | | | | EL | | | | | |
|---|---|---|---|---|---|---|---|---|---|---|---|---|---|---|
| n | δ | σ² | HPD-NI1 | HPD-NI2 | ET-NI1 | ET-NI2 | GCI | MOVER | HPD-NI1 | HPD-NI2 | ET-NI1 | ET-NI2 | GCI | MOVER |
| 30 | 10% | 0.3 | 0.9358 | 0.9420 | 0.9256 | 0.9322 | 0.9966 | 0.9968 | 0.5181 | 0.5257 | 0.4922 | 0.4994 | **0.7926** | 0.8028 |
| | | 0.5 | 0.9186 | 0.9258 | 0.9060 | 0.9136 | 0.9916 | 0.9914 | 0.6449 | 0.6509 | 0.6126 | 0.6184 | **0.8484** | 0.8548 |
| | | 0.8 | 0.9266 | 0.9304 | 0.9124 | 0.9156 | 0.9738 | 0.9724 | 0.7915 | 0.7966 | 0.7519 | 0.7568 | **0.9320** | 0.9342 |
| | | 1.0 | 0.9310 | 0.9330 | 0.9160 | 0.9186 | 0.9642 | 0.9606 | 0.8858 | 0.8905 | 0.8415 | 0.8460 | **0.9958** | 0.9963 |
| | | 2.0 | 0.9196 | 0.9160 | 0.9088 | 0.9024 | 0.9238 | 0.9222 | 1.3163 | 1.3196 | 1.2505 | 1.2536 | 1.3710 | 1.3653 |
| | 30% | 0.3 | 0.9552 | 0.9616 | 0.9446 | 0.9534 | 0.9928 | 0.9938 | 0.7172 | 0.7290 | 0.6814 | 0.6925 | 0.9426 | **0.9395** |
| | | 0.5 | 0.9374 | 0.9446 | 0.9254 | 0.9316 | 0.9840 | 0.9830 | 0.8527 | 0.8629 | 0.8101 | 0.8198 | 1.0288 | **1.0232** |
| | | 0.8 | 0.9304 | 0.9370 | 0.9186 | 0.9238 | 0.9746 | 0.9726 | 1.0137 | 1.0220 | 0.9630 | 0.9709 | 1.1550 | **1.1451** |
| | | 1.0 | 0.9348 | 0.9352 | 0.9210 | 0.9238 | 0.9660 | 0.9632 | 1.1194 | 1.1278 | 1.0634 | 1.0714 | 1.2412 | **1.2292** |
| | | 2.0 | 0.9202 | 0.9144 | 0.9056 | 0.9030 | 0.9222 | 0.9174 | 1.5505 | 1.5568 | 1.4729 | 1.4789 | 1.6359 | 1.6182 |
| | 50% | 0.3 | 0.9490 | 0.9620 | 0.9398 | 0.9510 | 0.9846 | 0.9850 | 1.0057 | 1.0248 | 0.9554 | 0.9735 | 1.1897 | **1.1661** |
| | | 0.5 | 0.9420 | 0.9540 | 0.9294 | 0.9436 | 0.9788 | 0.9798 | 1.1843 | 1.2010 | 1.1251 | 1.1410 | 1.3465 | **1.3199** |
| | | 0.8 | 0.9446 | 0.9538 | 0.9310 | 0.9410 | 0.9790 | 0.9770 | 1.3801 | 1.3945 | 1.3111 | 1.3248 | 1.5224 | **1.4924** |
| | | 1.0 | 0.9468 | 0.9528 | 0.9344 | 0.9420 | 0.9732 | 0.9698 | 1.4945 | 1.5076 | 1.4198 | 1.4323 | 1.6316 | **1.5976** |
| | | 2.0 | 0.9440 | 0.9450 | 0.9312 | 0.9334 | 0.9566 | 0.9504 | 1.9594 | 1.9713 | 1.8614 | 1.8727 | 2.0981 | **2.0567** |
| 50 | 10% | 0.3 | 0.8518 | 0.8598 | 0.8376 | 0.8470 | 0.9972 | 0.9972 | 0.3721 | 0.3757 | 0.3535 | 0.3569 | **0.6789** | 0.6841 |
| | | 0.5 | 0.8692 | 0.8756 | 0.8512 | 0.8612 | 0.9904 | 0.9912 | 0.4601 | 0.4631 | 0.4371 | 0.4399 | **0.6570** | 0.6609 |
| | | 0.8 | 0.9262 | 0.9302 | 0.9120 | 0.9138 | 0.9740 | 0.9726 | 0.5918 | 0.5943 | 0.5622 | 0.5645 | **0.6850** | 0.6869 |
| | | 1.0 | 0.9300 | 0.9324 | 0.9172 | 0.9206 | 0.9514 | 0.9494 | 0.6783 | 0.6806 | 0.6444 | 0.6466 | **0.7358** | 0.7371 |
| | | 2.0 | 0.9144 | 0.9138 | 0.9008 | 0.9012 | 0.9196 | 0.9176 | 1.0444 | 1.0455 | 0.9922 | 0.9932 | 1.0627 | 1.0611 |
| | 30% | 0.3 | 0.9072 | 0.9142 | 0.8926 | 0.9018 | 0.9936 | 0.9932 | 0.5207 | 0.5263 | 0.4946 | 0.5000 | 0.7935 | **0.7927** |
| | | 0.5 | 0.9024 | 0.9098 | 0.8864 | 0.8958 | 0.9832 | 0.9838 | 0.6099 | 0.6151 | 0.5794 | 0.5843 | 0.8062 | **0.8039** |
| | | 0.8 | 0.9310 | 0.9328 | 0.9158 | 0.9210 | 0.9722 | 0.9712 | 0.7414 | 0.7456 | 0.7044 | 0.7083 | 0.8528 | **0.8493** |
| | | 1.0 | 0.9388 | 0.9384 | 0.9222 | 0.9220 | 0.9594 | 0.9542 | 0.8264 | 0.8300 | 0.7850 | 0.7885 | 0.9044 | **0.8997** |
| | | 2.0 | 0.9252 | 0.9208 | 0.9082 | 0.9080 | 0.9250 | 0.9230 | 1.2095 | 1.2132 | 1.1490 | 1.1526 | 1.2433 | 1.2362 |
| | 50% | 0.3 | 0.9290 | 0.9408 | 0.9160 | 0.9256 | 0.9890 | 0.9894 | 0.7254 | 0.7355 | 0.6891 | 0.6987 | 0.9587 | **0.9496** |
| | | 0.5 | 0.9250 | 0.9344 | 0.9116 | 0.9232 | 0.9856 | 0.9858 | 0.8317 | 0.8404 | 0.7902 | 0.7984 | 1.0115 | **1.0004** |
| | | 0.8 | 0.9258 | 0.9350 | 0.9144 | 0.9202 | 0.9696 | 0.9670 | 0.9737 | 0.9810 | 0.9250 | 0.9319 | 1.1021 | **1.0884** |
| | | 1.0 | 0.9398 | 0.9448 | 0.9268 | 0.9326 | 0.9742 | 0.9718 | 1.0624 | 1.0699 | 1.0093 | 1.0164 | 1.1687 | **1.1537** |
| | | 2.0 | 0.9236 | 0.9224 | 0.9134 | 0.9098 | 0.9242 | 0.9206 | 1.4647 | 1.4693 | 1.3914 | 1.3958 | 1.5291 | 1.5113 |
| 100 | 10% | 0.3 | 0.7382 | 0.7428 | 0.7216 | 0.7290 | 0.9984 | 0.9984 | 0.2496 | 0.2510 | 0.2371 | 0.2385 | **0.5141** | 0.5161 |
| | | 0.5 | 0.8800 | 0.8850 | 0.8628 | 0.8688 | 0.9864 | 0.9868 | 0.3222 | 0.3235 | 0.3061 | 0.3073 | **0.4356** | 0.4373 |
| | | 0.8 | 0.9444 | 0.9422 | 0.9318 | 0.9304 | 0.9620 | 0.9634 | 0.4194 | 0.4202 | 0.3984 | 0.3992 | **0.4589** | 0.4599 |
| | | 1.0 | 0.9410 | 0.9404 | 0.9288 | 0.9298 | 0.9488 | 0.9474 | 0.4836 | 0.4845 | 0.4594 | 0.4603 | **0.5046** | 0.5052 |
| | | 2.0 | 0.9212 | 0.9192 | 0.9074 | 0.9074 | 0.9256 | 0.9256 | 0.7599 | 0.7605 | 0.7219 | 0.7224 | 0.7665 | 0.7661 |
| | 30% | 0.3 | 0.7956 | 0.8052 | 0.7786 | 0.7864 | 0.9948 | 0.9946 | 0.3520 | 0.3540 | 0.3344 | 0.3363 | 0.6199 | **0.6202** |
| | | 0.5 | 0.8850 | 0.8920 | 0.8688 | 0.8752 | 0.9848 | 0.9854 | 0.4197 | 0.4217 | 0.3987 | 0.4006 | 0.5532 | **0.5528** |
| | | 0.8 | 0.9378 | 0.9394 | 0.9218 | 0.9256 | 0.9606 | 0.9606 | 0.5195 | 0.5213 | 0.4935 | 0.4953 | 0.5697 | **0.5686** |
| | | 1.0 | 0.9420 | 0.9426 | 0.9302 | 0.9292 | 0.9490 | 0.9478 | 0.5840 | 0.5853 | 0.5548 | 0.5560 | 0.6135 | **0.6120** |
| | | 2.0 | 0.9216 | 0.9186 | 0.9104 | 0.9074 | 0.9240 | 0.9246 | 0.8790 | 0.8797 | 0.8351 | 0.8357 | 0.8883 | 0.8866 |
| | 50% | 0.3 | 0.8594 | 0.8662 | 0.8428 | 0.8532 | 0.9902 | 0.9906 | 0.4898 | 0.4936 | 0.4653 | 0.4689 | 0.7544 | **0.7522** |
| | | 0.5 | 0.8926 | 0.9000 | 0.8736 | 0.8828 | 0.9840 | 0.9826 | 0.5560 | 0.5592 | 0.5282 | 0.5312 | 0.7139 | **0.7106** |
| | | 0.8 | 0.9334 | 0.9340 | 0.9178 | 0.9228 | 0.9606 | 0.9584 | 0.6611 | 0.6641 | 0.6281 | 0.6309 | 0.7347 | **0.7303** |
| | | 1.0 | 0.9440 | 0.9456 | 0.9326 | 0.9326 | 0.9570 | 0.9562 | 0.7355 | 0.7382 | 0.6987 | 0.7013 | 0.7822 | **0.7772** |
| | | 2.0 | 0.9228 | 0.9178 | 0.9066 | 0.9072 | 0.9218 | 0.9194 | 1.0536 | 1.0558 | 1.0010 | 1.0030 | 1.0715 | 1.0662 |

**Remark:** Boldface indicates the recommended method for each case.

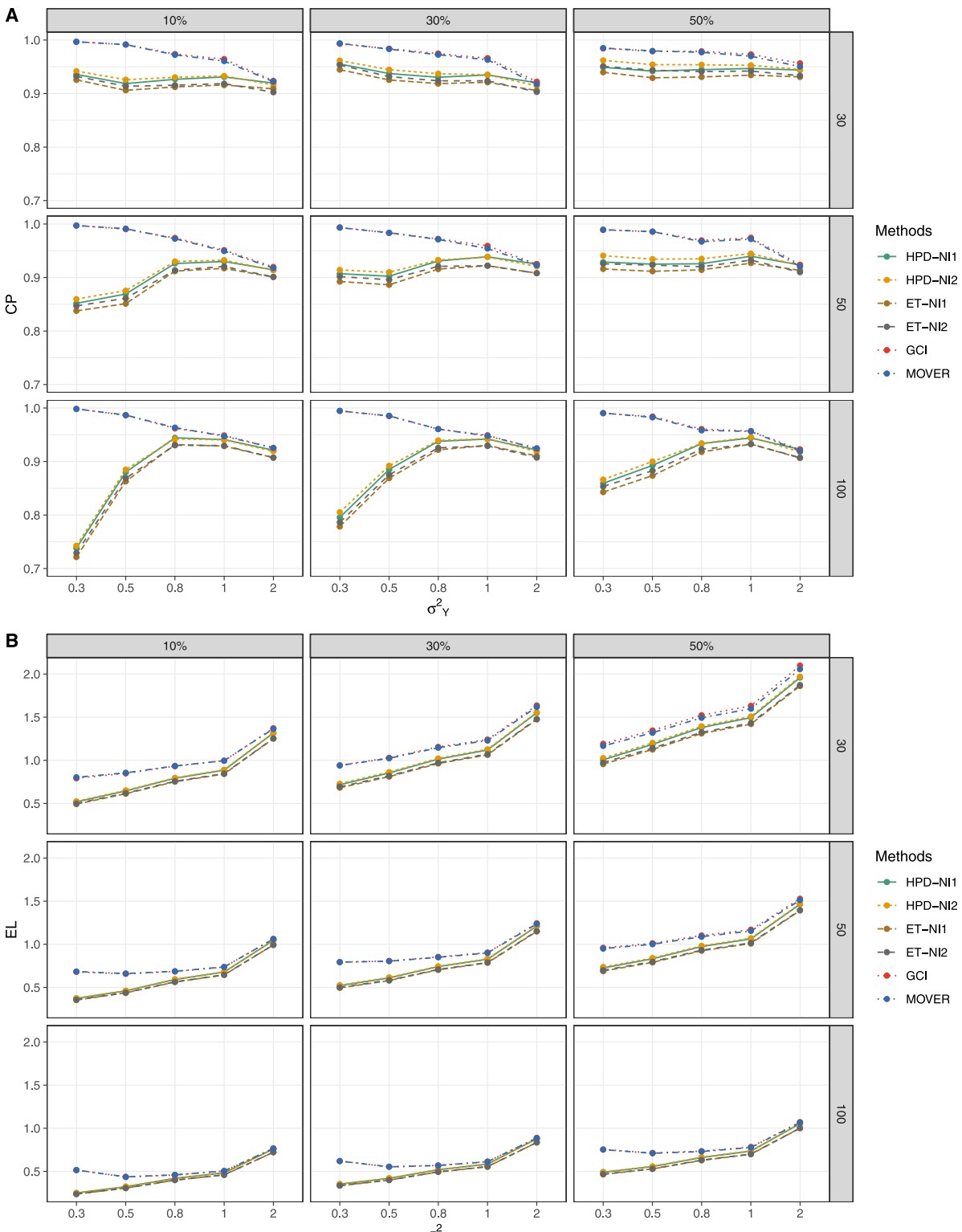

**Fig 1. Performance measures of 95%CIs for $\theta$: $a$ = 1 (A) Coverage probabilities and (B) Expected lengths.**

**Table 2. CP and EL performances of 95% CI for $\theta$: $a = 5$.**

| $a = 5$ | | | CP | | | | | | EL | | | | | |
|---|---|---|---|---|---|---|---|---|---|---|---|---|---|---|
| $n$ | $\delta$ | $\sigma^2$ | HPD-NI1 | HPD-NI2 | ET-NI1 | ET-NI2 | GCI | MOVER | HPD-NI1 | HPD-NI2 | ET-NI1 | ET-NI2 | GCI | MOVER |
| 30 | 10% | 0.3 | 0.9622 | 0.9634 | 0.9536 | 0.9544 | 0.9936 | 0.9942 | **0.4043** | 0.4138 | 0.3841 | 0.3931 | 0.6484 | 0.6627 |
| | | 0.5 | 0.9540 | 0.9546 | 0.9434 | 0.9424 | 0.9900 | 0.9894 | **0.4996** | 0.5077 | 0.4746 | 0.4823 | 0.6733 | 0.6857 |
| | | 0.8 | 0.9482 | 0.9456 | 0.9360 | 0.9348 | 0.9732 | 0.9726 | **0.6316** | 0.6379 | 0.6000 | 0.6060 | 0.7336 | 0.7436 |
| | | 1.0 | 0.9422 | 0.9390 | 0.9310 | 0.9286 | 0.9628 | 0.9610 | **0.7196** | 0.7256 | 0.6836 | 0.6893 | 0.7920 | 0.8005 |
| | | 2.0 | 0.9026 | 0.8996 | 0.8888 | 0.8836 | 0.9154 | 0.9136 | 1.1200 | 1.1240 | 1.0640 | 1.0678 | 1.1547 | 1.1576 |
| | 30% | 0.3 | 0.9556 | 0.9626 | 0.9458 | 0.9534 | 0.9896 | 0.9904 | **0.6013** | 0.6153 | 0.5713 | 0.5845 | 0.8073 | 0.8056 |
| | | 0.5 | 0.9564 | 0.9608 | 0.9462 | 0.9516 | 0.9802 | 0.9812 | **0.6989** | 0.7107 | 0.6639 | 0.6752 | 0.8578 | 0.8554 |
| | | 0.8 | 0.9576 | 0.9582 | 0.9466 | 0.9466 | 0.9722 | 0.9714 | **0.8309** | 0.8415 | 0.7894 | 0.7995 | 0.9430 | 0.9393 |
| | | 1.0 | 0.9558 | 0.9538 | 0.9444 | 0.9434 | 0.9660 | 0.9646 | **0.9294** | 0.9397 | 0.8830 | 0.8928 | 1.0188 | 1.0139 |
| | | 2.0 | 0.9136 | 0.9094 | 0.8990 | 0.8946 | 0.9222 | 0.9148 | 1.3422 | 1.3503 | 1.2751 | 1.2827 | 1.3995 | 1.3906 |
| | 50% | 0.3 | 0.9412 | 0.9558 | 0.9300 | 0.9440 | 0.9828 | 0.9850 | **0.8606** | 0.8825 | 0.8176 | 0.8384 | 1.0440 | 1.0217 |
| | | 0.5 | 0.9550 | 0.9634 | 0.9426 | 0.9566 | 0.9788 | 0.9792 | **0.9784** | 0.9987 | 0.9295 | 0.9487 | 1.1286 | 1.1065 |
| | | 0.8 | 0.9644 | 0.9680 | 0.9532 | 0.9592 | 0.9760 | 0.9738 | **1.1517** | 1.1695 | 1.0941 | 1.1111 | 1.2771 | 1.2531 |
| | | 1.0 | 0.9596 | 0.9614 | 0.9464 | 0.9524 | 0.9694 | 0.9662 | **1.2491** | 1.2658 | 1.1867 | 1.2025 | 1.3668 | 1.3415 |
| | | 2.0 | 0.9400 | 0.9380 | 0.9262 | 0.9240 | 0.9456 | 0.9382 | 1.7240 | 1.7386 | 1.6378 | 1.6516 | 1.8271 | 1.7974 |
| 50 | 10% | 0.3 | 0.9584 | 0.9576 | 0.9482 | 0.9460 | 0.9972 | 0.9982 | **0.2984** | 0.3030 | 0.2835 | 0.2879 | 0.5477 | 0.5545 |
| | | 0.5 | 0.9488 | 0.9480 | 0.9346 | 0.9378 | 0.9888 | 0.9886 | **0.3665** | 0.3705 | 0.3482 | 0.3519 | 0.5066 | 0.5131 |
| | | 0.8 | 0.9438 | 0.9420 | 0.9292 | 0.9304 | 0.9700 | 0.9692 | **0.4735** | 0.4767 | 0.4498 | 0.4529 | 0.5317 | 0.5374 |
| | | 1.0 | 0.9340 | 0.9322 | 0.9198 | 0.9156 | 0.9500 | 0.9474 | 0.5441 | 0.5467 | 0.5169 | 0.5193 | **0.5791** | 0.5842 |
| | | 2.0 | 0.9098 | 0.9068 | 0.8960 | 0.8924 | 0.9208 | 0.9206 | 0.8822 | 0.8840 | 0.8381 | 0.8398 | 0.8978 | 0.8995 |
| | 30% | 0.3 | 0.9466 | 0.9510 | 0.9356 | 0.9390 | 0.9902 | 0.9906 | **0.4478** | 0.4544 | 0.4254 | 0.4317 | 0.6723 | 0.6721 |
| | | 0.5 | 0.9498 | 0.9526 | 0.9400 | 0.9414 | 0.9800 | 0.9820 | **0.5117** | 0.5180 | 0.4861 | 0.4921 | 0.6618 | 0.6613 |
| | | 0.8 | 0.9488 | 0.9482 | 0.9372 | 0.9356 | 0.9708 | 0.9686 | **0.6172** | 0.6225 | 0.5864 | 0.5914 | 0.6918 | 0.6907 |
| | | 1.0 | 0.9418 | 0.9390 | 0.9298 | 0.9270 | 0.9538 | 0.9508 | **0.6874** | 0.6918 | 0.6530 | 0.6572 | 0.7390 | 0.7376 |
| | | 2.0 | 0.9110 | 0.9096 | 0.8976 | 0.8930 | 0.9184 | 0.9146 | 1.0421 | 1.0454 | 0.9900 | 0.9932 | 1.0664 | 1.0633 |
| | 50% | 0.3 | 0.9462 | 0.9522 | 0.9326 | 0.9414 | 0.9838 | 0.9854 | **0.6370** | 0.6480 | 0.6052 | 0.6156 | 0.8395 | 0.8305 |
| | | 0.5 | 0.9466 | 0.9524 | 0.9364 | 0.9428 | 0.9772 | 0.9774 | **0.7110** | 0.7212 | 0.6755 | 0.6851 | 0.8625 | 0.8531 |
| | | 0.8 | 0.9588 | 0.9596 | 0.9468 | 0.9478 | 0.9730 | 0.9706 | **0.8252** | 0.8345 | 0.7839 | 0.7927 | 0.9241 | 0.9133 |
| | | 1.0 | 0.9524 | 0.9542 | 0.9424 | 0.9404 | 0.9636 | 0.9612 | **0.9074** | 0.9157 | 0.8620 | 0.8699 | 0.9857 | 0.9742 |
| | | 2.0 | 0.9152 | 0.9090 | 0.8990 | 0.8962 | 0.9180 | 0.9120 | 1.2812 | 1.2878 | 1.2171 | 1.2234 | 1.3264 | 1.3136 |
| 100 | 10% | 0.3 | 0.9448 | 0.9462 | 0.9328 | 0.9332 | 0.9970 | 0.9972 | **0.2046** | 0.2062 | 0.1944 | 0.1959 | 0.3803 | 0.3829 |
| | | 0.5 | 0.9500 | 0.9522 | 0.9362 | 0.9382 | 0.9874 | 0.9884 | **0.2546** | 0.2559 | 0.2419 | 0.2431 | 0.3289 | 0.3317 |
| | | 0.8 | 0.9404 | 0.9390 | 0.9262 | 0.9270 | 0.9596 | 0.9582 | **0.3310** | 0.3321 | 0.3144 | 0.3155 | 0.3564 | 0.3588 |
| | | 1.0 | 0.9400 | 0.9390 | 0.9294 | 0.9240 | 0.9494 | 0.9490 | 0.3814 | 0.3823 | 0.3623 | 0.3632 | **0.3958** | 0.3980 |
| | | 2.0 | 0.9134 | 0.9128 | 0.8982 | 0.8996 | 0.9214 | 0.9210 | 0.6400 | 0.6408 | 0.6080 | 0.6087 | 0.6468 | 0.6477 |
| | 30% | 0.3 | 0.9384 | 0.9406 | 0.9272 | 0.9278 | 0.9936 | 0.9932 | **0.3106** | 0.3130 | 0.2951 | 0.2973 | 0.4904 | 0.4907 |
| | | 0.5 | 0.9464 | 0.9458 | 0.9348 | 0.9354 | 0.9760 | 0.9762 | **0.3566** | 0.3589 | 0.3388 | 0.3410 | 0.4396 | 0.4397 |
| | | 0.8 | 0.9458 | 0.9480 | 0.9338 | 0.9336 | 0.9586 | 0.9582 | **0.4313** | 0.4330 | 0.4098 | 0.4113 | 0.4616 | 0.4615 |
| | | 1.0 | 0.9434 | 0.9436 | 0.9274 | 0.9272 | 0.9498 | 0.9498 | **0.4813** | 0.4831 | 0.4573 | 0.4589 | 0.5003 | 0.5002 |
| | | 2.0 | 0.9258 | 0.9256 | 0.9100 | 0.9124 | 0.9280 | 0.9272 | 0.7482 | 0.7494 | 0.7108 | 0.7119 | 0.7574 | 0.7571 |
| | 50% | 0.3 | 0.9410 | 0.9440 | 0.9302 | 0.9326 | 0.9840 | 0.9842 | **0.4431** | 0.4472 | 0.4209 | 0.4249 | 0.6315 | 0.6291 |
| | | 0.5 | 0.9452 | 0.9464 | 0.9344 | 0.9332 | 0.9724 | 0.9734 | **0.4891** | 0.4927 | 0.4646 | 0.4680 | 0.5970 | 0.5939 |
| | | 0.8 | 0.9432 | 0.9458 | 0.9274 | 0.9306 | 0.9600 | 0.9580 | **0.5692** | 0.5724 | 0.5408 | 0.5437 | 0.6143 | 0.6106 |
| | | 1.0 | 0.9492 | 0.9488 | 0.9354 | 0.9370 | 0.9542 | 0.9516 | **0.6217** | 0.6246 | 0.5906 | 0.5934 | 0.6518 | 0.6481 |
| | | 2.0 | 0.9220 | 0.9208 | 0.9120 | 0.9076 | 0.9230 | 0.9226 | 0.9109 | 0.9131 | 0.8654 | 0.8675 | 0.9258 | 0.9222 |

**Remark:** Boldface indicates the recommended method for each case.

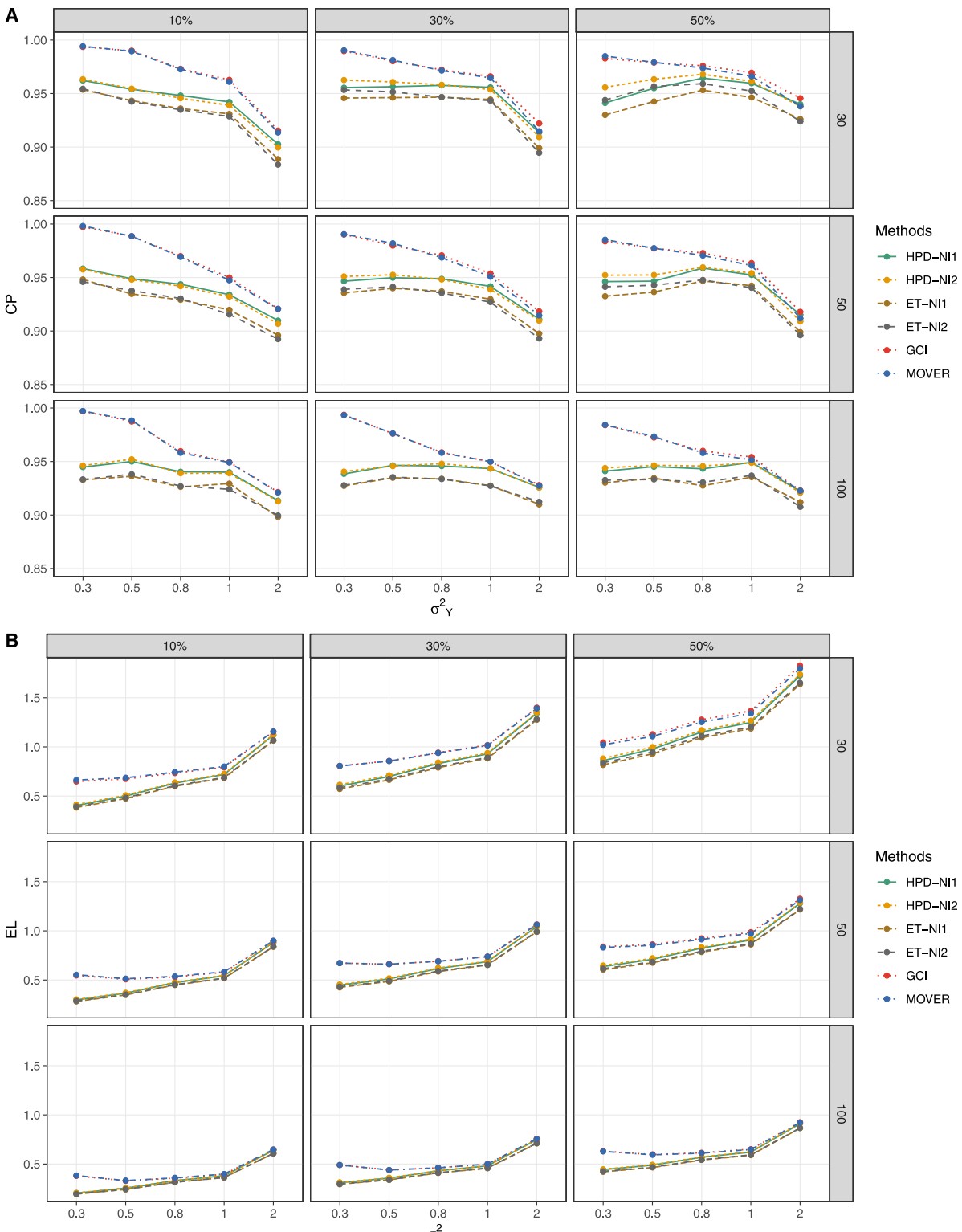

**Fig 2. Performance measures of 95%CIs for *θ*: *a* = 5 (A) Coverage probabilities and (B) Expected lengths.**

**Table 3. CP and EL performances of 95% CI for $\theta$: $a = 15$.**

| $a = 15$ | | | CP | | | | | | EL | | | | | |
|---|---|---|---|---|---|---|---|---|---|---|---|---|---|---|
| $n$ | $\delta$ | $\sigma^2$ | HPD-NI1 | HPD-NI2 | ET-NI1 | ET-NI2 | GCI | MOVER | HPD-NI1 | HPD-NI2 | ET-NI1 | ET-NI2 | GCI | MOVER |
| 30 | 10% | 0.3 | 0.9596 | 0.9670 | 0.9480 | 0.9586 | 0.9908 | 0.9936 | **0.3033** | 0.3151 | 0.2881 | 0.2994 | 0.4881 | 0.5065 |
| | | 0.5 | 0.9610 | 0.9620 | 0.9486 | 0.9510 | 0.9822 | 0.9876 | **0.3554** | 0.3660 | 0.3376 | 0.3477 | 0.4786 | 0.4978 |
| | | 0.8 | 0.9530 | 0.9524 | 0.9410 | 0.9394 | 0.9690 | 0.9708 | **0.4430** | 0.4524 | 0.4209 | 0.4298 | 0.5051 | 0.5248 |
| | | 1.0 | 0.9504 | 0.9460 | 0.9378 | 0.9344 | 0.9622 | 0.9614 | **0.5077** | 0.5162 | 0.4823 | 0.4904 | 0.5512 | 0.5704 |
| | | 2.0 | 0.9102 | 0.9020 | 0.8934 | 0.8892 | 0.9206 | 0.9168 | 0.8417 | 0.8477 | 0.7996 | 0.8053 | 0.8738 | 0.8878 |
| | 30% | 0.3 | 0.9390 | 0.9488 | 0.9300 | 0.9388 | 0.9752 | 0.9818 | **0.4983** | 0.5140 | 0.4734 | 0.4883 | 0.6623 | 0.6609 |
| | | 0.5 | 0.9516 | 0.9594 | 0.9388 | 0.9498 | 0.9742 | 0.9806 | **0.5538** | 0.5688 | 0.5261 | 0.5403 | 0.6790 | 0.6782 |
| | | 0.8 | 0.9574 | 0.9604 | 0.9440 | 0.9492 | 0.9688 | 0.9698 | **0.6388** | 0.6523 | 0.6068 | 0.6197 | 0.7219 | 0.7228 |
| | | 1.0 | 0.9590 | 0.9578 | 0.9476 | 0.9466 | 0.9638 | 0.9634 | **0.7044** | 0.7174 | 0.6692 | 0.6816 | 0.7688 | 0.7698 |
| | | 2.0 | 0.9248 | 0.9190 | 0.9108 | 0.9020 | 0.9270 | 0.9218 | 1.0401 | 1.0499 | 0.9881 | 0.9974 | 1.0907 | 1.0926 |
| | 50% | 0.3 | 0.9410 | 0.9524 | 0.9282 | 0.9406 | 0.9754 | 0.9768 | **0.7395** | 0.7642 | 0.7026 | 0.7260 | 0.9009 | 0.8773 |
| | | 0.5 | 0.9430 | 0.9588 | 0.9336 | 0.9482 | 0.9752 | 0.9774 | **0.8024** | 0.8254 | 0.7623 | 0.7841 | 0.9380 | 0.9159 |
| | | 0.8 | 0.9534 | 0.9610 | 0.9432 | 0.9502 | 0.9696 | 0.9706 | **0.9092** | 0.9311 | 0.8638 | 0.8846 | 1.0202 | 0.9993 |
| | | 1.0 | 0.9626 | 0.9684 | 0.9510 | 0.9582 | 0.9714 | 0.9710 | **0.9787** | 1.0000 | 0.9297 | 0.9500 | 1.0801 | 1.0607 |
| | | 2.0 | 0.9464 | 0.9422 | 0.9358 | 0.9316 | 0.9424 | 0.9390 | **1.3712** | 1.3893 | 1.3026 | 1.3198 | 1.4720 | 1.4523 |
| 50 | 10% | 0.3 | 0.9504 | 0.9586 | 0.9388 | 0.9480 | 0.9936 | 0.9946 | **0.2266** | 0.2323 | 0.2152 | 0.2207 | 0.3767 | 0.3859 |
| | | 0.5 | 0.9518 | 0.9544 | 0.9398 | 0.9406 | 0.9792 | 0.9826 | **0.2666** | 0.2717 | 0.2533 | 0.2581 | 0.3443 | 0.3544 |
| | | 0.8 | 0.9498 | 0.9476 | 0.9380 | 0.9372 | 0.9656 | 0.9668 | **0.3342** | 0.3386 | 0.3175 | 0.3216 | 0.3650 | 0.3753 |
| | | 1.0 | 0.9456 | 0.9450 | 0.9346 | 0.9336 | 0.9554 | 0.9542 | **0.3832** | 0.3871 | 0.3641 | 0.3678 | 0.4022 | 0.4125 |
| | | 2.0 | 0.9104 | 0.9078 | 0.8996 | 0.8936 | 0.9194 | 0.9190 | 0.6450 | 0.6478 | 0.6127 | 0.6154 | 0.6601 | 0.6673 |
| | 30% | 0.3 | 0.9444 | 0.9494 | 0.9326 | 0.9376 | 0.9818 | 0.9844 | **0.3856** | 0.3933 | 0.3663 | 0.3736 | 0.5326 | 0.5322 |
| | | 0.5 | 0.9572 | 0.9574 | 0.9468 | 0.9478 | 0.9752 | 0.9766 | **0.4211** | 0.4284 | 0.4001 | 0.4069 | 0.5077 | 0.5075 |
| | | 0.8 | 0.9458 | 0.9464 | 0.9310 | 0.9352 | 0.9572 | 0.9556 | **0.4804** | 0.4869 | 0.4563 | 0.4626 | 0.5237 | 0.5245 |
| | | 1.0 | 0.9492 | 0.9496 | 0.9374 | 0.9368 | 0.9510 | 0.9516 | **0.5267** | 0.5329 | 0.5004 | 0.5062 | 0.5567 | 0.5581 |
| | | 2.0 | 0.9264 | 0.9232 | 0.9128 | 0.9104 | 0.9276 | 0.9254 | 0.7973 | 0.8021 | 0.7574 | 0.7620 | 0.8192 | 0.8213 |
| | 50% | 0.3 | 0.9430 | 0.9522 | 0.9332 | 0.9422 | 0.9742 | 0.9760 | **0.5699** | 0.5820 | 0.5414 | 0.5529 | 0.7210 | 0.7107 |
| | | 0.5 | 0.9492 | 0.9552 | 0.9400 | 0.9470 | 0.9728 | 0.9738 | **0.6060** | 0.6176 | 0.5757 | 0.5867 | 0.7115 | 0.7010 |
| | | 0.8 | 0.9458 | 0.9510 | 0.9334 | 0.9380 | 0.9592 | 0.9598 | **0.6728** | 0.6836 | 0.6392 | 0.6494 | 0.7396 | 0.7305 |
| | | 1.0 | 0.9524 | 0.9560 | 0.9412 | 0.9424 | 0.9594 | 0.9598 | **0.7215** | 0.7322 | 0.6854 | 0.6956 | 0.7753 | 0.7667 |
| | | 2.0 | 0.9370 | 0.9322 | 0.9236 | 0.9160 | 0.9322 | 0.9294 | **1.0160** | 1.0242 | 0.9652 | 0.9730 | 1.0558 | 1.0490 |
| 100 | 10% | 0.3 | 0.9462 | 0.9516 | 0.9344 | 0.9406 | 0.9922 | 0.9920 | **0.1587** | 0.1609 | 0.1508 | 0.1528 | 0.2448 | 0.2483 |
| | | 0.5 | 0.9494 | 0.9494 | 0.9350 | 0.9360 | 0.9770 | 0.9776 | **0.1856** | 0.1875 | 0.1764 | 0.1781 | 0.2219 | 0.2259 |
| | | 0.8 | 0.9466 | 0.9444 | 0.9338 | 0.9324 | 0.9552 | 0.9554 | **0.2314** | 0.2328 | 0.2198 | 0.2212 | 0.2444 | 0.2486 |
| | | 1.0 | 0.9448 | 0.9436 | 0.9304 | 0.9290 | 0.9488 | 0.9482 | **0.2649** | 0.2661 | 0.2516 | 0.2528 | 0.2726 | 0.2766 |
| | | 2.0 | 0.9228 | 0.9216 | 0.9080 | 0.9078 | 0.9288 | 0.9284 | 0.4653 | 0.4661 | 0.4420 | 0.4428 | 0.4717 | 0.4746 |
| | 30% | 0.3 | 0.9468 | 0.9488 | 0.9346 | 0.9370 | 0.9812 | 0.9840 | **0.2732** | 0.2761 | 0.2596 | 0.2623 | 0.3600 | 0.3600 |
| | | 0.5 | 0.9472 | 0.9520 | 0.9318 | 0.9356 | 0.9702 | 0.9702 | **0.2945** | 0.2973 | 0.2798 | 0.2824 | 0.3339 | 0.3340 |
| | | 0.8 | 0.9540 | 0.9558 | 0.9424 | 0.9446 | 0.9594 | 0.9608 | **0.3348** | 0.3372 | 0.3180 | 0.3203 | 0.3514 | 0.3520 |
| | | 1.0 | 0.9468 | 0.9474 | 0.9338 | 0.9340 | 0.9502 | 0.9484 | **0.3674** | 0.3695 | 0.3490 | 0.3510 | 0.3783 | 0.3792 |
| | | 2.0 | 0.9214 | 0.9204 | 0.9062 | 0.9076 | 0.9234 | 0.9222 | 0.5688 | 0.5707 | 0.5404 | 0.5421 | 0.5776 | 0.5788 |
| | 50% | 0.3 | 0.9446 | 0.9480 | 0.9332 | 0.9370 | 0.9764 | 0.9758 | **0.4052** | 0.4095 | 0.3849 | 0.3890 | 0.5117 | 0.5081 |
| | | 0.5 | 0.9472 | 0.9490 | 0.9324 | 0.9378 | 0.9636 | 0.9650 | **0.4265** | 0.4307 | 0.4052 | 0.4092 | 0.4790 | 0.4755 |
| | | 0.8 | 0.9404 | 0.9430 | 0.9294 | 0.9318 | 0.9498 | 0.9498 | **0.4670** | 0.4707 | 0.4436 | 0.4471 | 0.4926 | 0.4894 |
| | | 1.0 | 0.9462 | 0.9468 | 0.9330 | 0.9318 | 0.9492 | 0.9482 | **0.5015** | 0.5050 | 0.4764 | 0.4797 | 0.5202 | 0.5173 |
| | | 2.0 | 0.9378 | 0.9346 | 0.9224 | 0.9226 | 0.9360 | 0.9338 | **0.7187** | 0.7211 | 0.6828 | 0.6850 | 0.7323 | 0.7306 |

**Remark:** Boldface indicates the recommended method for each case.

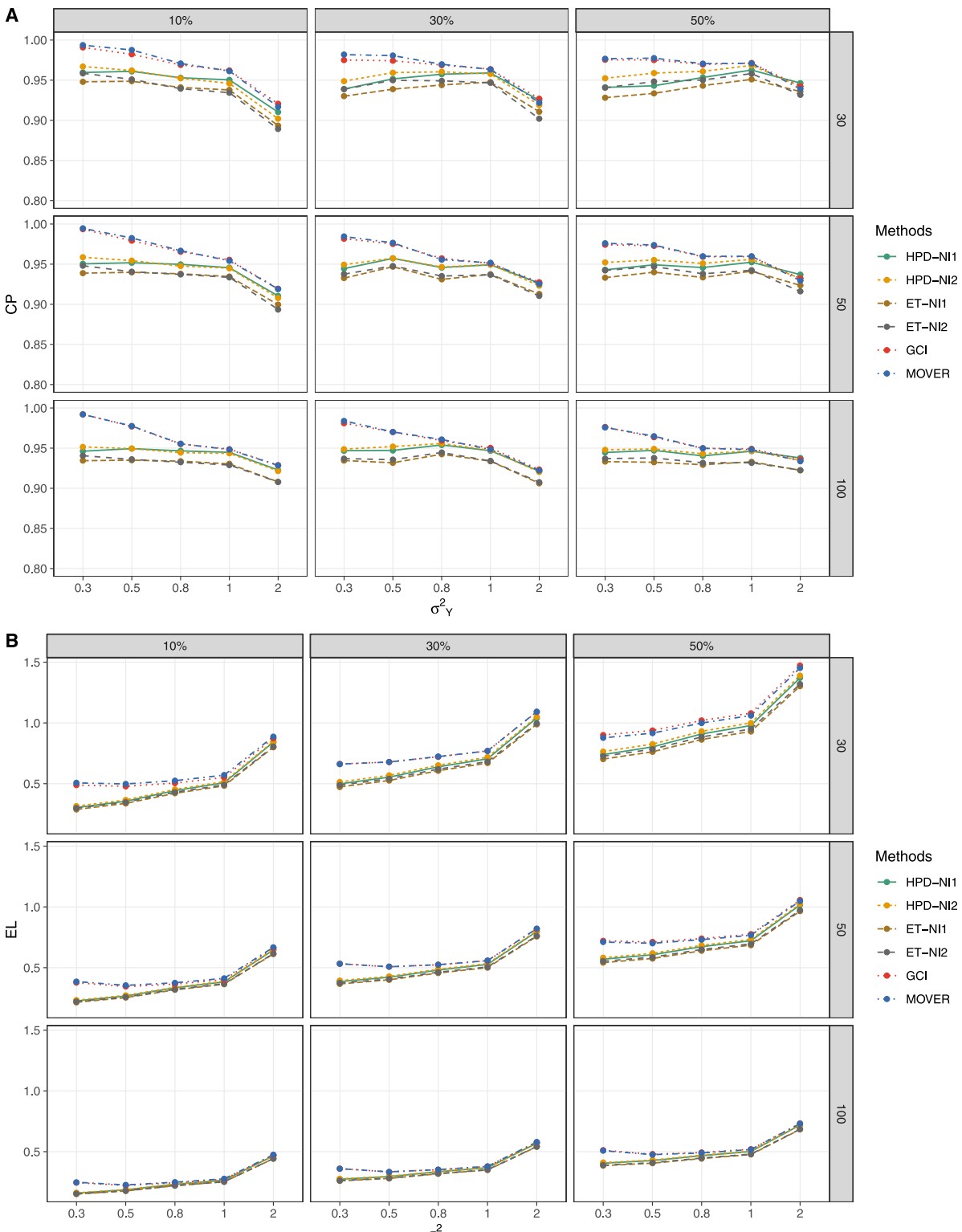

**Fig 3. Performance measures of 95%CIs for $\theta$: $a = 15$ (A) Coverage probabilities and (B) Expected lengths.**

**Table 4. Data on weekly natural rainfall in northern Thailand in the week 29 July to 4 August 2019.**

| Weekly natural rainfall data | | | | | | | | | | | | |
|---|---|---|---|---|---|---|---|---|---|---|---|---|
| 125.3 | 160.1 | 118.5 | 148.8 | 50 | 66.7 | 52.6 | 131.1 | 45.2 | 0 | 25.2 | 106.5 | 0 |
| 0 | 50.1 | 76.8 | 71.8 | 31.4 | 0 | 32.9 | 34.5 | 26.8 | 83.4 | 189.1 | 179.3 | 309.7 |
| 206.6 | 114.9 | 283.1 | 61.5 | 25 | 18 | 15 | 16.6 | 46 | 14.5 | 15 | 24.7 | 23 |
| 8.1 | 20.8 | 122.8 | 228.6 | 10.2 | 107.4 | 0 | 26.9 | 26.2 | 17.7 | 15.6 | 22.9 | 34.1 |
| 27 | 9.1 | 46.1 | 34.6 | 0 | 25.8 | 18.2 | 15.1 | 8.5 | 0 | | | |

Source: Thailand Meteorogical Deparment

URL: https://www.tmd.go.th/services/weekly_report.php

delta-three parameter lognormal distribution. The descriptive statistics for the data are as follows: $n = 62$, $\hat{a} = 1.7604$, $\delta = 11.29\%$, $\hat{\mu}_Y = 3.7256$ and $\hat{\sigma}_Y^2 = 1.0489$.

The mean of the weekly positive rainfall records is 62.5183 mm/wk. Computations of the 95% CIs for the BCIs, GCI, and MOVER for the estimated mean are reported in Table 6. The weekly rainfall amounts infer heavy rain (35.1–90.0 millimetre), as per the criteria of the TMD [31]. Importantly, it is in line with the TMD warnings of heavy downpours and flash floods to the population living in the at-risk areas. For $a = 1$, the evidence in support of the estimated CIs can be found in the simulation results in Section.

## Discussion

Random samples were drawn from data following a delta-three parameter lognormal distribution including zero observations of proportion $\delta$ and highly skewed non-zero values in the remaining proportion following a three-parameter lognormal distribution. This distribution offers a solution for how to handle highly skewed observed data that cannot be modeled using a two-parameter lognormal distribution. In this study, CI estimates for the mean of a delta-three parameter lognormal distribution were developed based on BCIs (HPD and ET-based NI intervals), GCI, and MOVER. Applying our proposed methods to predict the weekly natural rainfall amount was the motivation for this study.

When the threshold was large, HPD-NI1 provided better performance than the other methods in the extreme situation where the variance was small-to-medium, although it did not deal well with a large variance. The first reason is that the ET interval can substantially differ from the HPD region if the posterior density is highly skewed, as noted by Gelman *et al.* [20]. The next reason is that the NI1 prior was obtained from the prior of $\sigma_Y^2$ using its Fisher information matrix, which might make it stronger than the NI2 prior (the normal-gamma prior of $(\mu_Y, \sigma_Y^2)$). However, it is important to note the limitation of HPD-NI1 when dealing with large $\sigma_Y^2$. Likewise, the current study has a research gap in the perspectives on spatial information because it could be useful for statistical estimation if this study has been enabled by considerable insights associated with modeling framework using rainfall spatial analysis. See details in Banerjee *et al.* [32]. These need further research in the future.

## Conclusions

The present study aimed to propose BCIs-based NI1 and NI2 priors, GCI, and MOVER for the logarithm of the mean of delta-three parameter lognormal model. Our numerical evaluation shows that in situations of small threshold, MOVER maintained a good performance and obtained the recommended CIs for large proportion of zeros except for large variance, while the next recommended CIs were obtained by apply GCI. On the other hand, HPD-NI1

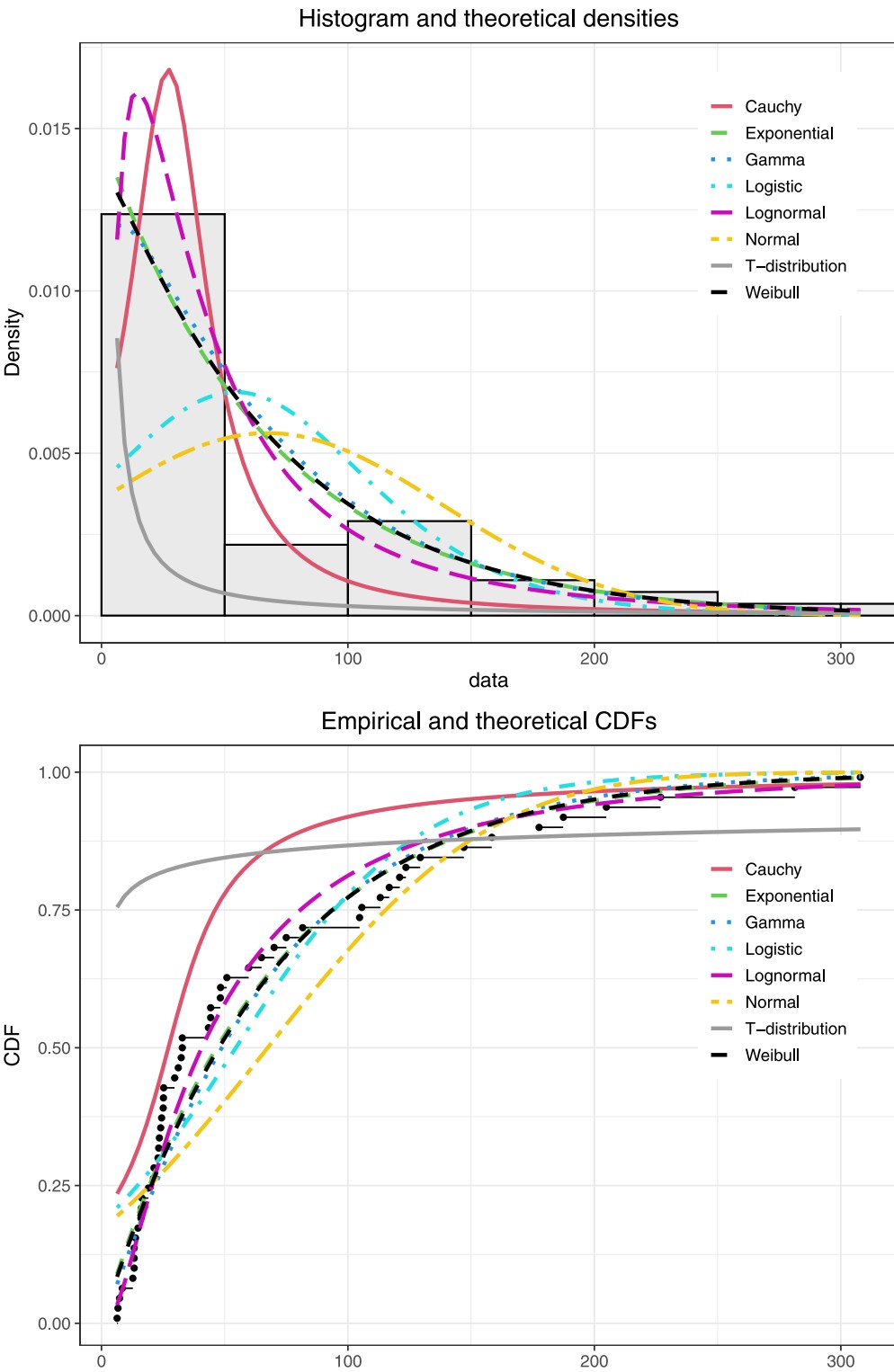

**Fig 4. Histogram and empirical CDF plots of weekly rainfall records in northern Thailand in the week 29 July to 4 August 2019.**

**Table 5. Results of AIC and BIC for weekly positive rainfall data.**

| Distributions | Criteria | |
|---|---|---|
| | AICs | BICs |
| Cauchy | 610.1506 | 614.1653 |
| Exponential | 575.2178 | 577.2251 |
| Gamma | 576.7389 | 580.7536 |
| Logistic | 622.7518 | 626.7665 |
| Lonormal | **569.3073** | **573.3219** |
| Normal | 628.9230 | 632.9376 |
| T-distribution | 612.0378 | 618.0598 |
| Weibull | 577.1703 | 581.1850 |

**Table 6. 95%CIs for the weekly average natural rainfall in northern Thailand.**

| Methods | 95% CIs for $\theta$ | | Lengths |
|---|---|---|---|
| | Lower | Upper | |
| HPD-NI1 | 44.9290 | 90.9399 | 46.0109 |
| HPD-NI2 | 44.8439 | 90.6832 | 45.8393 |
| ET-NI1 | 44.7487 | 90.6611 | 45.9124 |
| ET-NI2 | 43.3894 | 84.1893 | 40.7999 |
| GCI | 45.3618 | 91.5032 | 46.1414 |
| MOVER | 44.7933 | 91.6743 | 46.8810 |

performed quite well in situations of small-to-medium variance and a large threshold. There-fore, the HPD-NI1 is recommended for constructing CI estimation for the mean of a delta-three parameter lognormal distribution under these conditions. Furthermore, the GCI and MOVER are considered as the alternative methods.

## Supporting information

**S1 Abbreviations. Abbreviations commonly used throughout this article.**
(PDF)

**S1 Data. Data on weekly natural rainfall in northern Thailand in the week 29 July to 4 August 2019.**
(XLSX)

**S1 Fig. Performance measures of 95%CIs for $\theta$: $a$ = 1.** (A) Coverage probabilities and (B) Expected lengths.
(PDF)

**S2 Fig. Performance measures of 95%CIs for $\theta$: $a$ = 5.** (A) Coverage probabilities and (B) Expected lengths.
(PDF)

**S3 Fig. Performance measures of 95%CIs for $\theta$: $a$ = 15.** (A) Coverage probabilities and (B) Expected lengths.
(PDF)

**S4 Fig. Histogram and empirical CDF plots of weekly rainfall records in northern Thailand in the week 29 July to 4 August 2019.**
(PDF)

**S1 Table. CP and EL performances of 95% CI for $\theta$: $a$ = 1.**
(PDF)

**S2 Table. CP and EL performances of 95% CI for $\theta$: $a$ = 5.**
(PDF)

**S3 Table. CP and EL performances of 95% CI for $\theta$: $a$ = 15.**
(PDF)

**S4 Table. Data on weekly natural rainfall in northern Thailand in the week 29 July to 4 August 2019.**
(PDF)

**S5 Table. Results of AIC and BIC for weekly positive rainfall data.**
(PDF)

**S6 Table. 95%CIs for the weekly average natural rainfall in northern Thailand.**
(PDF)

## Acknowledgments

The authors would like to thank the academic editor and the reviewers for all useful and helpful comments on our manuscript. We would also like to thank Research and Development Institution, Uttaradit Rajabhat University for supporting the research management.

## Author Contributions

**Conceptualization:** Patcharee Maneerat, Sa-Aat Niwitpong.

**Data curation:** Patcharee Maneerat.

**Formal analysis:** Patcharee Maneerat.

**Funding acquisition:** Pisit Nakjai, Sa-Aat Niwitpong.

**Investigation:** Pisit Nakjai.

**Methodology:** Patcharee Maneerat, Sa-Aat Niwitpong.

**Project administration:** Pisit Nakjai.

**Resources:** Patcharee Maneerat.

**Software:** Pisit Nakjai.

**Supervision:** Sa-Aat Niwitpong.

**Visualization:** Patcharee Maneerat, Pisit Nakjai, Sa-Aat Niwitpong.

**Writing – original draft:** Patcharee Maneerat.

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
