## [Decision Letter · Decision Letter 0]

12 Jan 2022

PONE-D-21-29828Bayesian interval estimations for the mean of delta-three parameter lognormal distribution with application to heavy rainfall dataPLOS ONE

Dear Dr. Niwitpong,

Thank you for submitting your manuscript to PLOS ONE. After careful consideration, we feel that it has merit but does not fully meet PLOS ONE’s publication criteria as it currently stands. Therefore, we invite you to submit a revised version of the manuscript that addresses the points raised during the review process.

We look forward to receiving your revised manuscript.

Kind regards,

Inés P. Mariño, Ph.D.

Academic Editor

PLOS ONE

Journal Requirements:

Dr. Patcharee Maneerat and Dr. Pisit Nakjai were funded by

Thailand Science Research and Innovation. Dr. Sa-Aat Niwitpong was appreciated funding from King Mongkut’s University of Technology North Bangkok. Grant number: KMUTNB-65-KNOW-09.

The first and second authors were funded by

Thailand Science Research and Innovation. The third author was appreciated funding

from King Mongkut’s University of Technology North Bangkok. Grant number:

KMUTNB-65-KNOW-09.

Dr. Patcharee Maneerat and Dr. Pisit Nakjai were funded by

Thailand Science Research and Innovation. Dr. Sa-Aat Niwitpong was appreciated funding from King Mongkut’s University of Technology North Bangkok. Grant number: KMUTNB-65-KNOW-09.

Reviewers' comments:

Reviewer's Responses to Questions

**Comments to the Author**

1. Is the manuscript technically sound, and do the data support the conclusions?

Reviewer #1: Yes

2. Has the statistical analysis been performed appropriately and rigorously? 

Reviewer #1: Yes

3. Have the authors made all data underlying the findings in their manuscript fully available?

Reviewer #1: Yes

4. Is the manuscript presented in an intelligible fashion and written in standard English?

Reviewer #1: Yes

5. Review Comments to the Author

Reviewer #1: I enjoyed reading the manuscript. Bayesian inference certainly has several advantages over classical inference and the use of Bayesian confidence intervals to analyze rainfall is appealing. The manuscript is overall well written but I have some suggestions and recommendations that should situate the paper better in terms of its scientific merits.

1. While the authors have painstakingly derived analytical formulas for the Bayesian credible intervals, these could be derived using sampling-based inference by drawing posterior samples using MCMC. Software frameworks such as WinBUGS and STAN and RJAGS can easily handle such methods.

2. The one aspect of this research that has a gap is the lack of use of spatial information. A modeling framework that uses spatial analysis of rainfall will be veryuseful. There are a number of resources for Bayesian spatial analysis. The book,

Banerjee, S., Carlin, B.P. and Gelfand, A.E. (2014). Hierarchical Modeling and Analysis for Spatial Data. Second Edition. Taylor and Francis CRC.

This book should be referred.

6. PLOS authors have the option to publish the peer review history of their article (what does this mean?). If published, this will include your full peer review and any attached files.

Reviewer #1: No

---

## [Author Response · Author response to Decision Letter 0]

17 Mar 2022

Response to Reviewers

Journal: PLOS ONE

Manuscript ID: PONE-D-21-29828

Title name: Bayesian interval estimations for the mean of delta-three parameter lognormal distribution with application to heavy rainfall data 

Authors: Patcharee Maneerat, Pisit Nakjai and Sa-Aat Niwitpong

Dear Academic Editor and Reviewers: 

We would like to thank you for your time and patience in reviewing our manuscript and making insightful comments toward improving it for resubmission. The following are our responses to each of your comments.

Academic Editor” comments

1. We note that your manuscript is not formatted using one of PLOS ONE’s accepted file types. Please reattach your manuscript as one of the following file types: .doc, .docx, .rtf, or .tex (accompanied by a .pdf).

If your submission was prepared in LaTex, please submit your manuscript file in PDF format and attach your .tex file as “other.”

Response: The revised submission is prepared in LaTex and attached .tex file as “other.”

2. Please upload a Response to Reviewers letter which should include a point by point response to each of the points made by the Editor and / or Reviewers. (This should be uploaded as a 'Response to Reviewers' file type.) Please follow this link for more information: http://blogs.PLOS.org/everyone/2011/05/10/how-to-submit-your-revised-manuscript/

Response: Yes, we have uploaded a Response to Reviewers letter which should include a point by point response to each of the points made by the Editor and / or Reviewers.

Response: We have stated "The funders had no role in study design, data collection and analysis, decision to publish, or preparation of the manuscript." In the cover letter.

Reviewers' comments:

Reviewer #1: 

 I enjoyed reading the manuscript. Bayesian inference certainly has several advantages over classical inference and the use of Bayesian confidence intervals to analyze rainfall is appealing. The manuscript is overall well written, but I have some suggestions and recommendations that should situate the paper better in terms of its scientific merits.

1. While the authors have painstakingly derived analytical formulas for the Bayesian credible intervals, these could be derived using sampling-based inference by drawing posterior samples using MCMC. Software frameworks such as WinBUGS and STAN and RJAGS can easily handle such methods.

Response: Thank you for your valuable comments. We are looking forward to use these software frameworks in the future study. In this studies, we have used only package R version 4.1.3, 

in Algorithm 1 : BCIs (page 8), drawing posterior samples from parameters of the mean of delta-three parameter lognormal distribution can be approximately derived analytically, see pages 5-9. So we understand that MCMC is required when the posterior cannot be computed analytically. 

Moreover, it is also known form https://www.quora.com/Why-do-we-use-MCMC-algorithm-in-Bayesian-estimation that “Regarding Markov Chain, one of its bottleneck is the slower mixing time (or too many steps) to reach stationary distribution which usually worsen with an increasing number of dimensionality of the problem. Moreover, even when MCMC reaches stationary distribution, it still can only approximate the target distribution (to a normalising constant). In other words, the samples are not perfect.” So, drawing posterior samples using MCMC may be take longer times than the posterior samples in Algorithm 1: BCIs (page 8).

2. The one aspect of this research that has a gap is the lack of use of spatial information. A modeling framework that uses spatial analysis of rainfall will be very useful. There are a number of resources for Bayesian spatial analysis. The book,

Banerjee, S., Carlin, B.P. and Gelfand, A.E. (2014). Hierarchical Modeling and Analysis for Spatial Data. Second Edition. Taylor and Francis CRC.

This book should be referred.

Response: Thank you for your suggestions. This book is referred our manuscript in the Discussion section.

We look forward to hearing from you in the near future.

Sincerely yours,

Patcharee Maneerat, Pisit Nakjai, and Sa-Aat Niwitpong

The authors

---

## [Editor Report · Decision Letter 1]

22 Mar 2022

Bayesian interval estimations for the mean of delta-three parameter lognormal distribution with application to heavy rainfall data

PONE-D-21-29828R1

Dear Dr. Niwitpong,

We’re pleased to inform you that your manuscript has been judged scientifically suitable for publication and will be formally accepted for publication once it meets all outstanding technical requirements.

Kind regards,

Inés P. Mariño, Ph.D.

Academic Editor

PLOS ONE
---

## [Editor Report · Acceptance letter]

24 Mar 2022

PONE-D-21-29828R1 

Bayesian interval estimations for the mean of delta-three parameter lognormal distribution with application to heavy rainfall data 

Dear Dr. Niwitpong:

I'm pleased to inform you that your manuscript has been deemed suitable for publication in PLOS ONE. Congratulations! Your manuscript is now with our production department. 

Kind regards, 

on behalf of

Dr. Inés P. Mariño 

Academic Editor

PLOS ONE